# ADDIS: an adaptive discarding algorithm for online FDR control with conservative nulls

**Jinjin Tian**
Department of Statistics and Data Science
Carnegie Mellon University
Pittsburgh, PA 15213
jinjint@andrew.cmu.edu

**Aaditya Ramdas**
Department of Statistics and Data Science
Carnegie Mellon University
Pittsburgh, PA 15213
aramdas@cmu.edu

## Abstract

Major internet companies routinely perform tens of thousands of A/B tests each year. Such large-scale sequential experimentation has resulted in a recent spurt of new algorithms that can provably control the false discovery rate (FDR) in a fully online fashion. However, current state-of-the-art adaptive algorithms can suffer from a significant loss in power if null $p$-values are conservative (stochastically larger than the uniform distribution), a situation that occurs frequently in practice. In this work, we introduce a new adaptive discarding method called ADDIS that provably controls the FDR and achieves the best of both worlds: it enjoys appreciable power increase over all existing methods if nulls are conservative (the practical case), and rarely loses power if nulls are exactly uniformly distributed (the ideal case). We provide several practical insights on robust choices of tuning parameters, and extend the idea to asynchronous and offline settings as well.

## 1 Introduction

Rapid data collection is making the online testing of hypotheses increasingly essential, where a stream of hypotheses $H_1, H_2, \ldots$ is tested sequentially one by one. On observing the data for the $t$-th test which is usually summarized as a $p$-value $P_t$, and without knowing the outcomes of the future tests, we must make the decision of whether to reject the corresponding null hypothesis $H_t$ (thus proclaiming a "discovery"). Typically, a decision takes the form $I(P_t \leq \alpha_t)$ for some $\alpha_t \in (0, 1)$, meaning that we reject the null hypothesis when the $p$-value is smaller than some threshold $\alpha_t$. An incorrectly rejected null hypothesis is called a false discovery. Let $\mathcal{R}(T)$ represent the set of rejected null hypotheses until time $T$, and $\mathcal{H}_0$ be the unknown set of true null hypotheses; then, $\mathcal{R}(T) \cap \mathcal{H}_0$ is the set of false discoveries. Then some natural error metrics are the false discovery rate (FDR), modified FDR (mFDR) and power, which are defined as

$$\text{FDR}(T) \equiv \mathbb{E}\left[\frac{|\mathcal{H}_0 \cap \mathcal{R}(T)|}{|\mathcal{R}(T)| \vee 1}\right], \quad \text{mFDR}(T) \equiv \frac{\mathbb{E}\left[|\mathcal{H}_0 \cap \mathcal{R}(T)|\right]}{\mathbb{E}\left[|\mathcal{R}(T)| \vee 1\right]}, \quad \text{power} \equiv \mathbb{E}\left[\frac{|\mathcal{H}_0^c \cap \mathcal{R}(T)|}{|\mathcal{H}_0^c|}\right].$$
(1)

The typical aim is to maximize power, while have $\text{FDR}(T) \leq \alpha$ at any time $T \in \mathbb{N}$, for some prespecified constant $\alpha \in (0, 1)$. It is well known that setting every $\alpha_t \equiv \alpha$ does not provide any control of the FDR in general. Indeed, the FDR can be as large as one in this case, see [1, Section 1] for an example. This motivates the need for special methods for online FDR control (that is, for determining $\alpha_t$ in an online manner).

**Past work.** Foster and Stine [2] proposed the first "alpha-investing" (AI) algorithm for online FDR control, which was later extended to the generalized alpha-investing methods (GAI) by Aharoni and Rosset [3]. A particularly powerful GAI algorithm called LORD was proposed by Javanmard and

Montanari [4]. Soon after, Ramdas et al. [1] proposed a modification called LORD++ that uniformly improved the power of LORD. Most recently, Ramdas et al. [5] developed the "adaptive" SAFFRON algorithm, and alpha-investing is shown to be a special case of the more general SAFFRON framework. SAFFRON arguably represents the state-of-the-art, achieving significant power gains over all other algorithms including LORD++ in a range of experiments.

However, an important point is that SAFFRON is more powerful only when the $p$-values are exactly uniformly distributed under the null hypothesis. In practice, one frequently encounters *conservative* nulls (see below), and in this case SAFFRON can have lower power than LORD++ (see Figure 1).

**Uniformly conservative nulls.** When performing hypothesis testing, we always assume that the $p$-value $P$ is *valid*, which means that if the null hypothesis is true, we have $\Pr\{P \le x\} \le x$ for all $x \in [0, 1]$. Ideally, a $p$-value is exactly uniformly distributed, which means that the inequality holds with equality. However, we say a null $p$-value is *conservative* if the inequality is strict, and often the nulls are *uniformly conservative*, which means that under the null hypothesis, we have

$$\Pr\{P/\tau \le x \mid P \le \tau\} \le x \quad \text{for all } x, \tau \in (0, 1). \tag{2}$$

As an obvious first example, the $p$-values being exactly uniform (the ideal setting) is a special case. Indeed, for a uniform $U \sim U[0, 1]$, if you know that $U$ is less than (say) $\tau = 0.4$, then the conditional distribution of $U$ is just $U[0, 0.4]$, which means that $U/0.4$ has a uniform distribution on $[0, 1]$, and hence $\Pr\{U/0.4 \le x \mid U \le 0.4\} \le x$ for any $x \in (0, 1)$. A mathematically equivalent definition of uniformly conservative nulls is that the CDF $F$ of a null $p$-value $P$ satisfies the following property:

$$F(\tau x) \le x F(\tau), \quad \text{for all } 0 \le x, \tau \le 1. \tag{3}$$

Hence, any null $p$-value with convex CDF is uniformly conservative. Particularly, when $F$ is differentiable, the convexity of $F$ is equivalent to its density $f$ being monotonically increasing. Here are two tangible examples of tests with uniformly conservative nulls:

- A test of Gaussian mean: we test the null hypothesis $H_0 : \mu \le 0$ against the alternative $H_1 : \mu > 0$; the observation is $Z \sim N(\mu, 1)$ and the $p$-value is computed as $P = \Phi(-Z)$, where $\Phi$ is the standard Gaussian CDF.

- A test of Gaussian variance: we observe $Z \sim N(0, \sigma)$ and we wish to test the null hypothesis $H_0 : \sigma \le 1$ against the $H_1 : \sigma > 1$ and the $p$-value is $P = 2\Phi(-|Z|)$.

It is easy to verify that, if the true $\mu$ in the first test is strictly smaller than zero, or the true $\sigma$ in the second test is strictly smaller than one, then the corresponding null $p$-values have monotonically increasing density, thus being uniformly conservative. More generally, Zhao et al. [6] presented the following sufficient condition for a one-dimensional exponential family with true parameter $\theta$: when the true $\theta$ is strictly smaller than $\theta_0$, the uniformly most powerful (UMP) test of $H_0 : \theta \le \theta_0$ versus $H_1 : \theta > \theta_0$ is uniformly conservative. Since the true underlying state of nature is rarely *exactly* at the boundary of the null set (like $\mu = 0$ or $\sigma = 1$ or $\theta = \theta_0$ in the above examples), it is common in practice to encounter uniformly conservative nulls. In the context of A/B testing, this corresponds to testing $H_0 : \mu_B \le \mu_A$ against $H_1 : \mu_B > \mu_A$, when in reality, $B$ (the new idea) is strictly worse than $A$ (the existing system), a very likely scenario.

**Our contribution** The main contribution of this paper is a new method called ADDIS (an ADaptive algorithm that DIScards conservative nulls), that compensates for the power loss of SAFFRON with conservative nulls. ADDIS is based on a new serial estimate of the false discovery proportion, having adaptivity to both fraction of nulls (like SAFFRON) and the conservativeness of nulls (unlike SAFFRON). As shown in Figure 1, ADDIS enjoys appreciable power increase over SAFFRON as well as LORD++ under settings with many conservative nulls, and rarely loses power when the nulls are exactly uniformly distributed (not conservative). Our work is motivated by recent work by Zhao et al. [6] and Ellis et al. [7] who study nonadaptive offline multiple testing problems with conservative nulls, and ADDIS can be regarded as extending their work to both online and adaptive settings. The connection to the offline setting is that ADDIS effectively employs a "discarding" rule, which states we should discard (that is, not test) a hypothesis with $p$-value exceeding certain threshold. Beyond the online setting, we also incorporate this rule into several other existing FDR methods, and formally prove that the resulting new methods still control the FDR, while demonstrating numerically they have a consistent power advantage over the original methods. Figure 2 presents the relational chart of

historical FDR control methods together with some of the new methods we proposed. As far as we know, we provide the first method that adapts to the conservativeness of nulls in the online setting.

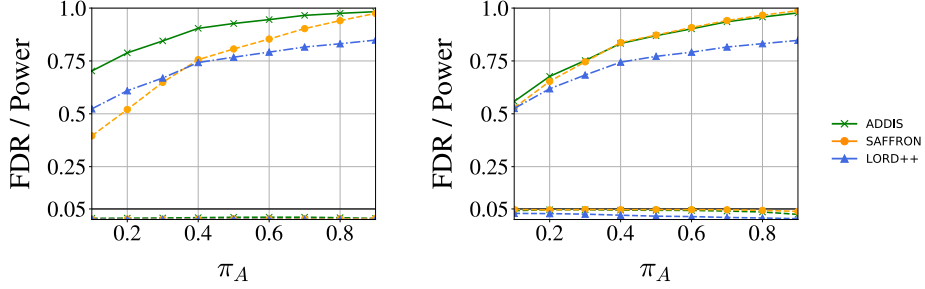

Figure 1: Statistical power and FDR versus fraction of non-null hypotheses $\pi_A$ for ADDIS, SAFFRON and LORD++ at target FDR level $\alpha = 0.05$ (solid black line). The curves above 0.05 line display the power of each methods versus $\pi_A$, while the lines below 0.05 display the FDR of each methods versus $\pi_A$. The experimental setting is described in Section 3: we set $\mu_A = 3$ for both figures, but $\mu_N = -1$ for the left figure and $\mu_N = 0$ for the right figure (hence the left nulls are conservative, the right nulls are not). These figures show that (a) all considered methods do control the FDR at level 0.05, (b) SAFFRON sometimes loses its superiority over its nonadaptive variant LORD++ with conservative nulls (i.e. $\mu_N < 0$); and (c) ADDIS is more powerful than SAFFRON and LORD++ with conservative nulls, while loses almost nothing under settings with uniform nulls (i.e. $\mu_N = 0$).

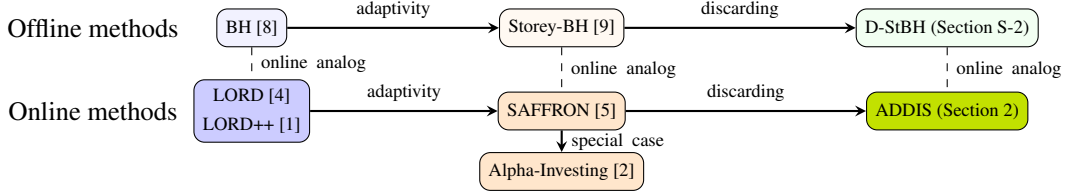

Figure 2: Historical context: ADDIS generalizes SAFFRON, which generalizes Alpha-Investing and LORD++. Analogously, D-StBH (supplement) generalizes Storey-BH, which generalizes BH.

**Paper outline.** In Section 2, we derive the ADDIS algorithm and state its guarantees (FDR and mFDR control), deferring proofs to the supplement. Specifically, in Section 2.4, we discuss how to choose the hyperparameters in ADDIS to balance adaptivity and discarding for optimal power. Section 3 shows simulations which demonstrate the advantage of ADDIS over non-discarding or non-adaptive methods. We then generalize the "discarding" rule of ADDIS in Section 4 and use it to obtain the "discarding" version of many other methods under various settings. We also show the error control with formal proofs for those variants in the supplement. Finally, we present a short summary in Section 5. The code to reproduce all figures in the paper is included in the supplement.

## 2 The ADDIS algorithm

Before deriving the ADDIS algorithm, it is useful to set up some notation. Recall that $P_j$ is the $p$-value for testing hypothesis $H_j$. For some sequences $\{\alpha_t\}_{t=1}^{\infty}$, $\{\tau_t\}_{t=1}^{\infty}$ and $\{\lambda_t\}_{t=1}^{\infty}$, where each term is in the range $[0, 1]$, define the indicator random variables

$$S_j = \mathbf{1}\{P_j \leq \tau_j\}, \quad C_j = \mathbf{1}\{P_j \leq \lambda_j\}, \quad R_j = \mathbf{1}\{P_j \leq \alpha_j\}.$$

They respectively answer the questions: "*was $H_j$ selected for testing? (or was it discarded?)*", "*was $H_j$ a candidate for rejection?*" and "*was $H_j$ rejected, yielding a discovery?*". We call the sets

$$S(t) = \{j \in [t] : S_j = 1\}, \quad C(t) = \{j \in [t] : C_j = 1\}, \quad R(t) = \{j \in [t] : R_j = 1\}$$

as the "selected (not discarded) set", "candidate set" and "rejection set" after $t$ steps respectively. Similarly, we define $R_{1:t} = \{R_1, \ldots, R_t\}$, $C_{1:t} = \{C_1, \ldots, C_t\}$ and $S_{1:t} = \{S_1, \ldots, S_t\}$. In what follows in this section and the next section, we repeatedly encounter the filtration

$$\mathcal{F}^t := \sigma(R_{1:t}, C_{1:t}, S_{1:t}).$$

We insist that $\alpha_t$, $\lambda_t$ and $\tau_t$ are predictable, that is they are measurable with respect to $\mathcal{F}^{t-1}$. This means that $\alpha_t, \lambda_t, \tau_t$ are really mappings from $\{R_{1:t-1}, C_{1:t-1}, S_{1:t-1}\} \mapsto [0,1]$.

The presentation is cleanest if we assume that the $p$-values from the different hypotheses are independent (which would be the case if each A/B test was based on fresh data, for example). However, we can also prove mFDR control under a mild form of dependence: we call the null $p$-values *conditionally uniformly conservative* if for any $t \in \mathcal{H}_0$, we have that

$$\forall x, \tau \in (0,1), \ \Pr\{P_t/\tau \leq x \mid P_t \leq \tau, \mathcal{F}^{t-1}\} \leq x. \tag{4}$$

Note that the above condition is equivalent to the (marginally) uniformly conservative property (2) if the $p$-values are independent, and hence $P_t$ is independent of $\mathcal{F}^{t-1}$. For simplicity, we will refer this "conditionally uniformly conservative" property still as "uniformly conservative".

## 2.1 Deriving ADDIS algorithm

Denote the (unknown) false discovery proportion by $\text{FDP} \equiv \frac{|\mathcal{H}_0 \cap \mathcal{R}(T)|}{|\mathcal{R}(T)| \vee 1}$. As mentioned in [5], one can control the FDR at any time $t$ by instead controlling an oracle estimate of the FDP, given by

$$\text{FDP}^*(t) := \frac{\sum_{j \leq t, j \in \mathcal{H}_0} \alpha_j}{|R(t)| \vee 1}. \tag{5}$$

This means that if we can keep $\text{FDP}^*(t) \leq \alpha$ at all times $t$, then we can prove that $\text{FDR}(t) \leq \alpha$ at all times $t$. Since the set of nulls $\mathcal{H}_0$ is unknown, LORD++ [1] is based on the simple upper bound of $\text{FDP}^*(t)$, defined as $\widehat{\text{FDP}}_{\text{LORD++}}(t)$, and SAFFRON [5] is based on a more nuanced adaptive bound on $\text{FDP}^*(t)$, defined as $\widehat{\text{FDP}}_{\text{SAFFRON}}(t)$, obtained by choosing a predictable sequence $\{\lambda_j\}_{j=1}^{\infty}$; where

$$\widehat{\text{FDP}}_{\text{LORD++}}(t) := \frac{\sum_{j \leq t} \alpha_j}{|R(t)| \vee 1}, \quad \widehat{\text{FDP}}_{\text{SAFFRON}}(t) := \frac{\sum_{j \leq t} \alpha_j \frac{\mathbf{1}\{P_j > \lambda_j\}}{1 - \lambda_j}}{|R(t)| \vee 1}. \tag{6}$$

It is easy to fix $\alpha_1 < \alpha$, and then update $\alpha_2, \alpha_3, \ldots$ in an online fashion to maintain the invariant $\widehat{\text{FDP}}_{\text{LORD++}}(t) \leq \alpha$ at all times, which the authors prove suffices for FDR control, while it is also proved that keeping $\widehat{\text{FDP}}_{\text{SAFFRON}}(t) \leq \alpha$ at all times suffices for FDR control at any time. However, we expect $\widehat{\text{FDP}}_{\text{SAFFRON}}(t)$ to be closer [1] to $\text{FDP}^*(t)$ than $\widehat{\text{FDP}}_{\text{LORD++}}(t)$, and since SAFFRON better uses its FDR budget, it is usually more powerful than LORD++. SAFFRON is called an "adaptive" algorithm, because it is the online analog of the Storey-BH procedure [9], which adapts to the proportion of nulls in the offline setting.

However, in the case when there are many conservative null $p$-values (whose distribution is stochastically larger than uniform), many terms in $\{\frac{\mathbf{1}\{\lambda_j < P_j\}}{1 - \lambda_j} : j \in \mathcal{H}_0\}$ may have expectations much larger than one, making $\widehat{\text{FDP}}_{\text{SAFFRON}}(t)$ an overly conservative estimator of $\text{FDP}^*(t)$, and thus causing a loss in power. In order to fix this, we propose a new empirical estimator of $\text{FDP}^*(t)$. We pick two predictable sequences $\{\lambda_j\}_{j=1}^{\infty}$ and $\{\tau_j\}_{j=1}^{\infty}$ such that $\lambda_j < \tau_j$ for all $j$, for example setting $\lambda_j = 1/4, \tau_j = 1/2$ for all $j$, and define

$$\widehat{\text{FDP}}_{\text{ADDIS}}(t) := \frac{\sum_{j \leq t} \alpha_j \frac{\mathbf{1}\{\lambda_j < P_j \leq \tau_j\}}{\tau_j - \lambda_j}}{|R(t)| \vee 1} \quad \underset{\theta_j := \frac{\lambda_j}{\tau_j}}{\equiv} \quad \frac{\sum_{j \leq t} \alpha_j \frac{\mathbf{1}\{P_j \leq \tau_j\} \mathbf{1}\{P_j/\tau_j > \theta_j\}}{\tau_j(1 - \theta_j)}}{|R(t)| \vee 1} \tag{7}$$

With many conservative nulls, the claim that ADDIS is more powerful than SAFFRON, is based on the idea that the numerator of $\widehat{\text{FDP}}_{\text{ADDIS}}(t)$ is a much tighter estimator of $\sum_{j \leq t, j \in \mathcal{H}_0} \alpha_j$, compared with that of $\widehat{\text{FDP}}_{\text{SAFFRON}}(t)$. In order to see why this is true, we provide the following lemma.

**Lemma 1.** *If a null $p$-value $P$ has a differentiable convex CDF, then for any constants $a, b \in (0,1)$, we have*

$$\frac{\Pr\{ab < P \leq b\}}{b(1-a)} \leq \frac{\Pr\{P > a\}}{(1-a)}. \tag{8}$$

The proof of Lemma 1 is presented in Section S-4. Recalling definition (3), Lemma 1 implies that for some uniformly conservative nulls, our estimator $\widehat{\text{FDP}}_{\text{ADDIS}}$ will be tighter than $\widehat{\text{FDP}}_{\text{SAFFRON}}$ in expectation, and thus an algorithm based on keeping $\widehat{\text{FDP}}_{\text{ADDIS}} \leq \alpha$ is expected to have higher power.

**ADDIS algorithm**  We now present the general ADDIS algorithm. Given user-defined sequences $\{\lambda_j\}_{j=1}^{\infty}$ and $\{\tau_j\}_{j=1}^{\infty}$ as described previously, we call an online FDR algorithm as an instance of the "ADDIS algorithm" if it updates $\alpha_t$ in a way such that it maintains the invariant $\widehat{\text{FDP}}_{\text{ADDIS}}(t) \leq \alpha$. We also enforce the constraint that $\tau_t > \lambda_t \geq \alpha_t$ for all $t$, which is needed for correctness of the proof of FDR control. This is not a major restriction since we often choose $\alpha = 0.05$, and the algorithms set $\alpha_t \leq \alpha$, in which case $\tau_t > \lambda_t \geq 0.05$ easily satisfies the needed constraint. Now, the main nontrivial question is how to ensure the invariant in a fully online fashion. We address this by providing an explicit instance of ADDIS algorithm, called ADDIS* (Algorithm 1), in the following section. From the form of the invariant $\widehat{\text{FDP}}_{\text{ADDIS}}(t) \leq \alpha$, we observe that any $p$-value $P_j$ that is bigger than $\tau_j$ has no influence on the invariant, as if it never existed in the sequence at all. This reveals that ADDIS effectively implements a "discarding" rule: it discards $p$-values exceeded a certain threshold. If the $p$-value is not discarded, then $P_j/\tau_j$ is a valid $p$-value and we resort to using adaptivity like (6).

## 2.2  ADDIS*: an instance of ADDIS algorithm using constant $\lambda$ and $\tau$

Here we present an instance of ADDIS algorithm, with choice of $\lambda_j \equiv \lambda$ and $\tau_j \equiv \tau$ for all $j$. (We consider constant $\lambda$ and $\tau$ for simplicity, but these can be replaced by $\lambda_j$ and $\tau_j$ at time $j$.)

---

**Algorithm 1:** The ADDIS* algorithm

---

**Input:** FDR level $\alpha$, discarding threshold $\tau \in (0,1]$, candidate threshold $\lambda \in [0,\tau)$, sequence $\{\gamma_j\}_{j=0}^{\infty}$ which is nonnegative, nonincreasing and sums to one, initial wealth $W_0 \leq \alpha$.

**for** $t = 1, 2, \ldots$ **do**

Reject the $t$-th null hypothesis if $P_t \leq \alpha_t$, where $\alpha_t := \min\{\lambda, \widehat{\alpha}_t\}$, and

$$\widehat{\alpha}_t := (\tau - \lambda)\left(W_0 \gamma_{S^t - C_{0+}} + (\alpha - W_0)\gamma_{S^t - \kappa_1^* - C_{1+}} + \alpha \sum_{j \geq 2} \gamma_{S^t - \kappa_j^* - C_{j+}}\right).$$

Here, $S^t = \sum_{i<t} \mathbf{1}\{P_i \leq \tau\}$, $\quad C_{j+} = \sum_{i=\kappa_j+1}^{t-1} \mathbf{1}\{P_i \leq \lambda\}$,

$\kappa_j = \min\{i \in [t-1] : \sum_{k \leq i} \mathbf{1}\{P_k \leq \alpha_k\} \geq j\}$, $\quad \kappa_j^* = \sum_{i \leq \kappa_j} \mathbf{1}\{P_i \leq \tau\}$.

**end**

---

In Section S-5.2, we verify that $\alpha_t$ is a monotonic function of the past[2]. In Section S-10, we present Algorithm S-3, which is an equivalent version of the above ADDIS* algorithm, but it explicitly discards $p$-values larger than $\tau$, thus justifying our use of the term "discarding" throughout this paper. Note that if we choose $\lambda \geq \alpha$, then the constraint $\alpha_t := \min\{\lambda, \widehat{\alpha}_t\}$ is vacuous and reduces to $\alpha_t := \widehat{\alpha}_t$, because $\widehat{\alpha}_t \leq \alpha$ by construction. The power of ADDIS varies with $\lambda$ and $\tau$, as discussed further in Section 2.4.

## 2.3  Error control of ADDIS algorithm

Here we present error control guarantees for ADDIS, and defer proofs to Section S-5 and Section S-6.

**Theorem 1.** *If the null $p$-values are uniformly conservative* (4)*, and suppose we choose $\alpha_j, \lambda_j$ and $\tau_j$ such that $\tau_j > \lambda_j \geq \alpha_j$ for each $j \in \mathbb{N}$, then we have:*

*(a) any algorithm with $\widehat{\text{FDP}}_{\text{ADDIS}}(t) \leq \alpha$ for all $t \in \mathbb{N}$ also enjoys* mFDR$(t) \leq \alpha$ *for all $t \in \mathbb{N}$. If we additionally assume that the null $p$-values are independent of each other and of the non-nulls, and always choose $\alpha_t, \lambda_t$ and $1 - \tau_t$ to be monotonic functions of the past for all $t$, then we additionally have:*

*(b) any algorithm with $\widehat{\text{FDP}}_{\text{ADDIS}}(t) \leq \alpha$ for all $t \in \mathbb{N}$ also enjoys* FDR$(t) \leq \alpha$ *for all $t \in \mathbb{N}$. As an immediate corollary, any ADDIS algorithm enjoys* mFDR *control, and ADDIS* (Algorithm 1) additionally enjoys* FDR *control since it is a monotonic rule.*

The above result only holds for nonrandom times. Below, we also show that any ADDIS algorithm controls mFDR at any stopping time with finite expectation.

**Theorem 2.** *Assume that the null p-values are uniformly conservative, and that $\min_j\{\tau_j - \lambda_j\} > \epsilon$ for some $\epsilon > 0$. Then, for any stopping time $T_{\text{stop}}$ with finite expectation, any algorithm that maintains the invariant $\widehat{\text{FDP}}_{\text{ADDIS}}(t) \leq \alpha$ for all $t$ enjoys $\text{mFDR}(T_{\text{stop}}) \leq \alpha$.*

Once more, the conditions for the theorem are not restrictive because the sequences $\{\lambda_j\}_{j=1}^{\infty}$ and $\{\tau_j\}_{j=1}^{\infty}$ are user-chosen, and $\lambda_j = 1/4, \tau_j = 1/2$ is a reasonable default choice, as we justify next.

### 2.4 Choosing $\tau$ and $\lambda$ to balance adaptivity and discarding

As we mentioned before, the power of our ADDIS* algorithm is closely related to the hyper-parameters $\lambda$ and $\tau$. In fact, there is also an interaction between the hyper-parameters $\lambda$ and $\tau$, which means that one cannot decouple the effect of each on power. One can see this interaction clearly in Figure 3 which displays a trade off between adaptivity ($\lambda$) and discarding ($\tau$). Indeed, the right sub-figure displays a "sweet spot" for choosing $\lambda, \tau$, which should neither be too large nor too small.

Ideally, one would hope that there exists some universally optimal choice of $\lambda, \tau$ that yields maximum power. Unfortunately, the relationship between power and these parameters changes with the underlying distribution of the null and alternate $p$-values, as well as their relative frequency. Therefore, below, we only provide a heuristic argument about how to tune these parameters for ADDIS*.

Recall that the ADDIS* algorithm is derived by tracking the empirical estimator $\widehat{\text{FDP}}_{\text{ADDIS}}$ (7) with fixed $\lambda$ and $\tau$, and keeping it bounded by $\alpha$ over time. Since $\widehat{\text{FDP}}_{\text{ADDIS}}$ serves as an estimate of the oracle FDP* (5), it is natural to expect higher power with a more refined (i.e. tighter) estimator $\widehat{\text{FDP}}_{\text{ADDIS}}$. One simple way to choose $\lambda$ and $\tau$ is to minimize the expectation of the indicator term in the estimator. Specifically, if the CDF of all $p$-values is $F$, then an oracle would choose $\lambda, \tau$ as

$$(\lambda^*, \tau^*) \in \arg\min_{\lambda < \tau \in (0,1)} \frac{F(\tau) - F(\lambda)}{\tau - \lambda}. \tag{9}$$

In order to remove the constraints between the two variables, we again define $\theta = \lambda/\tau$, then the optimization problem (9) is equivalent to

$$(\theta^*, \tau^*) \in \arg\min_{\theta, \tau \in (0,1)} \frac{F(\tau) - F(\theta\tau)}{\tau(1 - \theta)} \equiv (g \circ F)(\theta, \tau). \tag{10}$$

We provide some empirical evidence to show the quality of the above proposal. The left subfigure in Figure 3 shows the heatmap of $(g \circ F)$ and the right one shows the empirical power of ADDIS* with $p$-values generate from $F$ versus different $\theta$ and $\tau$ (the left is simply evaluating a function, the right requires repeated simulation). The same pattern is consistent across other reasonable choices of $F$, as shown in Section S-11. We can see that the two subfigures in Figure 3 show basically the same pattern, with similar optimal choices of parameters $\theta$ and $\tau$. Therefore, we suggest choosing $\lambda$ and $\tau$ as defined in (9), if prior knowledge of $F$ is available; otherwise it seems like $\theta \in [0.25, 0.75]$ and $\tau \in [0.15, 0.55]$ are safe choices, and for simplicity we use $\tau = \theta = 0.5$ as defaults, that is $\tau = 0.5, \lambda = 0.25$, in similar experimental settings. We leave the study of time-varying $\lambda_j$ and $\tau_j$ as future work.

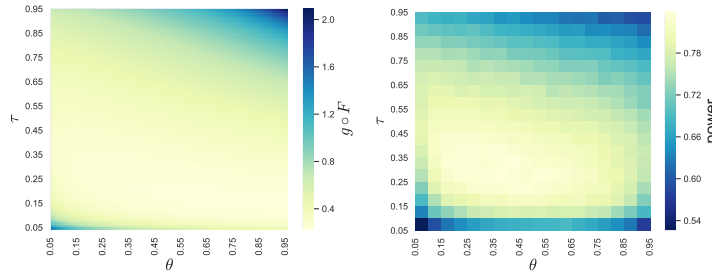

Figure 3: The left figure shows the heatmap of function $g \circ F$, where $F$ is the CDF of $p$-values drawn as described in Section 3 with $\mu_N = -1, \mu_A = 3, \pi_A = 0.2$. The right figure is the empirical power of ADDIS* versus different choice of $\theta$ and $\tau$, with $p$-values drawn from $F$. The figures are basically of the same pattern, with similar optimal values of $\theta$ and $\tau$.

## 3   Numerical experiments

In this section, we numerically compare the performance of ADDIS against the previous state-of-the-art algorithm SAFFRON [5], and other well-studied algorithms like LORD++ [4], LOND [10] and Alpha-investing [2]. Specifically, we use ADDIS* defined in Algorithm 1 as the representative of our ADDIS algorithm. Though as discussed in Section 2.4, there is no universally optimal constants, given the minimal nature of our assumptions, we will use some reasonable default choices in the numerical studies to have a glance at the advantage of ADDIS algorithm. The constants $\lambda = 0.25$, $\tau = 0.5$ and sequence $\{\gamma_j\}_{j=0}^{\infty}$ with $\gamma_j \propto 1/(j+1)^{-1.6}$ were found to be particularly successful, thus are our default choices for hyperparameters in ADDIS*. We choose the infinite constant sequence $\gamma_j \propto \frac{1}{(j+1)^{1.6}}$, and $\lambda = 0.5$ for SAFFRON, which yielded its best performance. We use $\gamma_j \propto \frac{\log((j+1)\wedge 2)}{(j+1)e^{\sqrt{\log(j+1)}}}$ for LORD++ and LOND, which is shown to maximize its power in the Gaussian setting [4]. The proportionality constant of $\{\gamma_j\}_{i=0}^{\infty}$ is determined so that the sequence $\{\gamma_j\}_{i=0}^{\infty}$ sums to one.

We consider the standard experimental setup of testing Gaussian means, with $M = 1000$ hypotheses. More precisely, for each index $i \in \{1, 2, \ldots, M\}$, the null hypotheses take the form $H_i : \mu_i \leq 0$, which are being tested against the alternative $H_{iA} : \mu_i > 0$. The observations are independent Gaussians $Z_i \sim N(\mu_i, 1)$, where $\mu_i \equiv \mu_N \leq 0$ with probability $1 - \pi_A$ and $\mu_i \equiv \mu_A > 0$ with probability $\pi_A$. The one-sided $p$-values are computed as $P_i = \Phi(-Z_i)$, which are uniformly conservative if $\mu_N < 0$ as discussed in the introduction (and the lower $\mu_N$ is, the more conservative the $p$-value). In the rest of this section, for each algorithm, we use target FDR $\alpha = 0.05$ and estimate the empirical FDR and power by averaging over 200 independent trials. Figure 4 shows that ADDIS has higher power than all other algorithms when the nulls are conservative (i.e. $\mu_N < 0$), and ADDIS matches the power of SAFFRON without conservative nulls (i.e. $\mu_N = 0$).

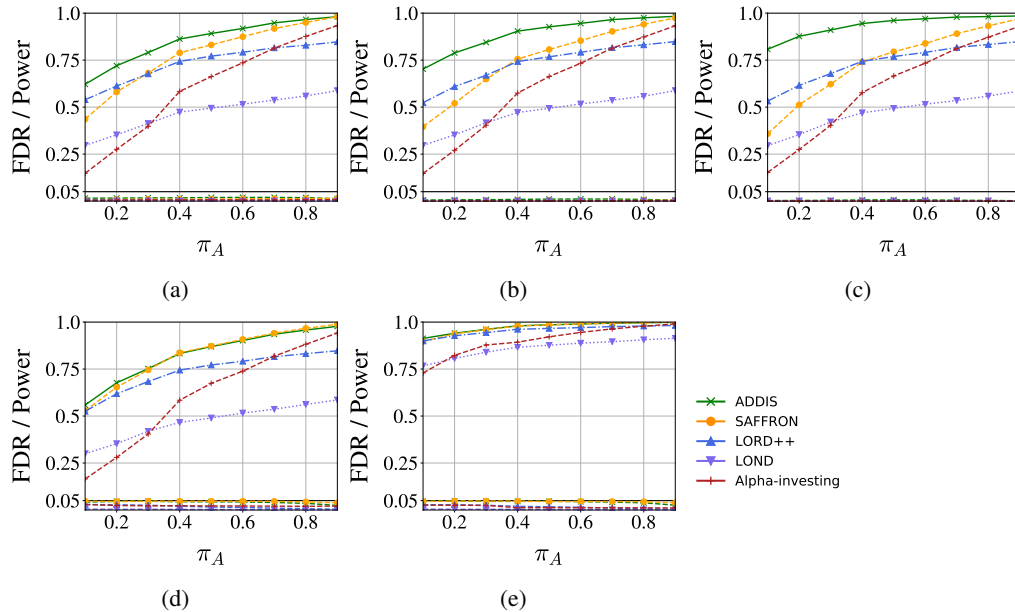

(a)    (b)    (c)

(d)    (e)

Figure 4: Statistical power and FDR versus fraction of non-null hypotheses $\pi_A$ for ADDIS, SAFFRON, LORD++, LOND, and Alpha-investing at target FDR level $\alpha = 0.05$ (solid black line). The lines above the solid black line are the power of each methods versus $\pi_A$, and the lines below are the FDR of each methods versus $\pi_A$. The $p$-values are drawn using the Gaussian model as described in the text, while we set $\mu_N = -0.5$ in plot (a), $\mu_N = -1$ in plot (b), $\mu_N = -1.5$ in plot (c), and $\mu_N = 0$ in plots (d) and (e). And we set $\mu_A = 3$ in plots (a, b, c, d), $\mu_A = 4$ in plot (e). These plots show that (1) FDR is under control for all methods in all settings; (2) ADDIS enjoys appreciable power increase as compared to all the other four methods; (3) the more conservative the nulls are (the more negative $\mu_N$ is), the more significant the power increase of ADDIS is; (4) ADDIS matches SAFFRON and remains the best in the setting with uniform (not conservative) nulls.

# 4 Generalization of the discarding rule

As we discussed before in Section 2, one way to interpret what ADDIS is doing is that it is "discarding" the large $p$-values. We say ADDIS may be regarded as applying the "discarding" rule to SAFFRON. Naturally, we would like to see whether the general advantage of this simple rule can be applied to other FDR control methods, and under more complex settings. We present the following generalizations and leave the details (formal setup, proofs) to supplement for interested readers.

- **Extension 1: non-adaptive methods with discarding**
  We derive the discarding version of LORD++ , which we would refer as D-LORD, in Section S-1, with proved FDR control.

- **Extension 2: discarding with asynchronous $p$-values**
  In a recent preprint, Zrnic et al. [11] show how to generalize existing online FDR control methods to what they call the asynchronous multiple testing setting. They consider a doubly-sequential setup, where one is running a sequence of sequential experiments, many of which could be running in parallel, starting and ending at different times arbitrarily. In Section S-3, we show how to unite the discarding rule from this paper with the "principle of pessimism" of Zrnic et al. [11] to derive even more powerful asynchronous online FDR algorithms, which we would refer as ADDIS$_{\text{async}}$.

- **Extension 3: Offline FDR control with discarding**
  In Section S-2, we provide a new offline FDR control method called D-StBH, to show how to incorporate the discarding rule with the Storey-BH method, which is a common offline adaptive testing procedure [12, 13]. Note that in the offline setting, the discarding rule is fundamentally the same as the idea of [6, 7], which were only applied to non-adaptive multiple testing.

The following simulation results in Figure 5, which are plotted in the same format as in Section 3, show that those discarding variants (marked with green color) enjoys the same type of advantage over their non-discarding counterparts: they are consistently more powerful under settings with many conservative nulls and do not lose much power under settings without conservative nulls.

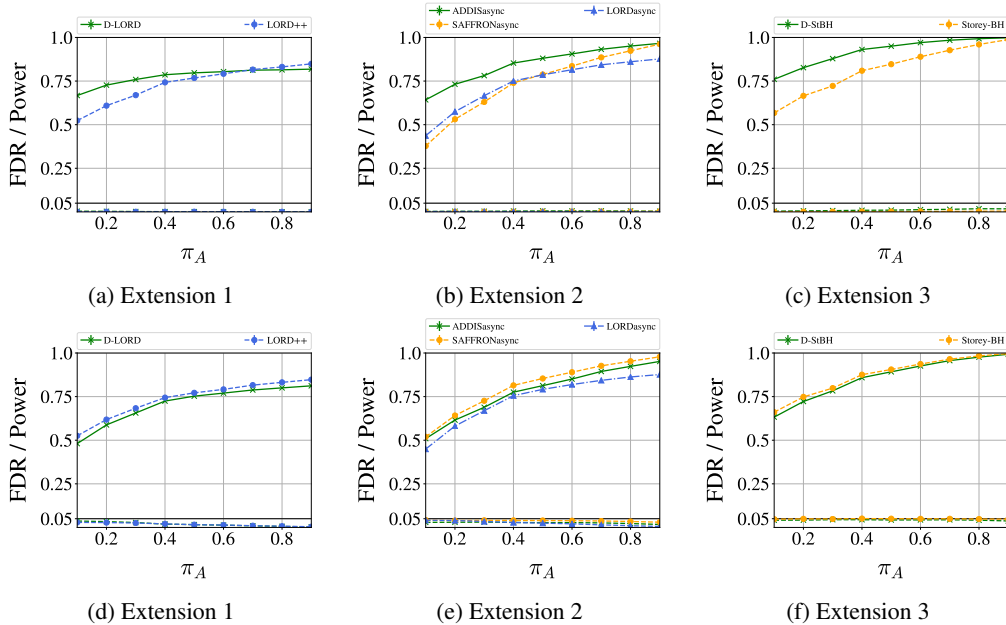

Figure 5: Statistical power and FDR versus fraction of non-null hypotheses $\pi_A$ for extended methods mentioned above at target FDR level $\alpha = 0.05$ (solid black line). The $p$-values are drawn using the Gaussian model as described in the text, while we set $\mu_A = 3$ for all the figures, but $\mu_N = -1$ in plots (a, b, c), $\mu_N = 0$ in plots (d, e, f). We additionally set the finish time for the $j$-th test as $E_j \sim j - 1 + \text{Geom}(0.5)$ in plots (b, e), which means the duration time of each individual tests independently follows Geometric distribution with succeed probability 0.5.

# 5  Conclusion

In this work, we propose a new online FDR control method, ADDIS, to compensate for the unnecessary power loss of current online FDR control methods due to conservative nulls. Numerical studies show that ADDIS is significantly more powerful than current state of arts, under settings with many conservative nulls, and rarely lose power under settings without conservative nulls. We also discuss the trade-off between adaptivity and discarding in ADDIS, together with some good heuristic of how to balance them to obtain higher power. In the end, we generalize the main idea of ADDIS to a simple but powerful rule "discarding", and incorporate the rule with many current online FDR control methods under various settings to generate corresponding more powerful variants. For now, we mainly examine the power advantage of ADDIS algorithm with constant $\lambda$ and $\tau$, though for future work, how to choose time varying $\{\lambda_j\}_{j=1}^{\infty}$ and $\{\tau_j\}_{j=1}^{\infty}$ in a data adaptive matter with provable power increase is worthy of more attention.

## Footnotes

[1]To see this intuitively, consider the case when (a) $\lambda_j \equiv 1/2$ for all $j$, (b) there is a significant fraction of non-nulls, and the non-null $p$-values are all smaller than 1/2 (strong signal), and (c) the null $p$-values are exactly uniformly distributed. Then, $\frac{\mathbf{1}\{1/2 < P_j\}}{1/2}$ evaluates to 0 for every non-null, and equals one for every null in expectation. Thus, in this case, $\mathbb{E}\left[\sum_{j \leq t} \alpha_j \frac{\mathbf{1}\{\lambda_j < P_j\}}{1 - \lambda_j}\right] = \mathbb{E}\left[\sum_{j \leq t, j \in \mathcal{H}_0} \alpha_j\right] \ll \mathbb{E}\left[\sum_{j \leq t} \alpha_j\right]$.

[2]We say that a function $f_t(R_{1:t-1}, C_{1:t-1}, S_{1:t-1}) : \{0,1\}^{3(t-1)} \to [0,1]$ is a monotonic function of the past, if $f_t$ is coordinatewise nondecreasing in $R_i$ and $C_i$, and is coordinatewise nonincreasing in $S_i$. This is a generalization of the monotonicity of SAFFRON [5], which is recovered by setting $S_i = 1$ for all $i$, that is we never discard any $p$-value.

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
