[Supplementary Material]

# Supplementary material for
# ADDIS: an adaptive discarding algorithm for online FDR control with conservative nulls

**Jinjin Tian**
Department of Statistics and Data Science
Carnegie Mellon University
Pittsburgh, PA 15213
jinjint@andrew.cmu.edu

**Aaditya Ramdas**
Department of Statistics and Data Science
Carnegie Mellon University
Pittsburgh, PA 15213
aramdas@cmu.edu

## S-1   D-LORD: LORD++ with discarding

Instead of applying discarding rule to LORD described in [1], we apply the discarding rule to its equivalent form LORD++ under the framework of GAI++ [2] for theoretical simplicity, and call the resulted variant as D-LORD. Now consider uniformly conservative $p$-values as defined in (4), where the filtration $\mathcal{F}^{t-1} = \sigma(R_{1:t-1}, S_{1:t-1})$. As before, we derive D-LORD from an empirical estimate of FDP* defined in (5). Specifically, let

$$\widehat{\text{FDP}}_{\text{D-LORD}}(t) := \frac{\sum_{j \leq t} \frac{\alpha_j}{\tau_j} \mathbf{1}\{P_j \leq \tau_j\}}{|R(t)| \vee 1}. \tag{S-1}$$

Compare $\widehat{\text{FDP}}_{\text{D-LORD}}$ with the original estimator that LORD++ based upon

$$\widehat{\text{FDP}}_{\text{LORD++}}(t) := \frac{\sum_{j \leq t} \alpha_j}{|R(t)| \vee 1}, \tag{S-2}$$

we say $\widehat{\text{FDP}}_{\text{D-LORD}}$ is a better estimator, since with many conservative null $p$-values, its numerator will be a much tighter estimate of $\sum_{j \leq t, j \in \mathcal{H}_0} \alpha_j$, compared with the naive estimate of LORD++ that is $\sum_{j \leq t} \alpha_j$. To see why this is true, just notice that the expectation of $\frac{\mathbf{1}\{P_j \leq \tau_j\}}{\tau_j}$ will be much smaller than 1 for conservative null $p$-values. We call an online FDR algorithm as an instance of the "D-LORD algorithm" if it updates $\alpha_t$ in a way such that it maintains the invariant $\widehat{\text{FDP}}_{\text{D-LORD}}(t) \leq \alpha$ for all $t$. We show how to ensure this invariant in a fully online fashion by providing an explicit instance of D-LORD with constant $\tau$ as the following D-LORD* algorithm. The simulation results in Section 4 demonstrate the power advantage of D-LORD* over LORD++.

---
**Algorithm S-1:** The D-LORD* algorithm

---
**Input:** FDR level $\alpha$, discarding threshold $\tau \in (0,1]$, sequence $\{\gamma_j\}_{j=0}^{\infty}$ which is nonnegative, nonincreasing and sums to one, initial wealth $W_0 \leq \alpha$.

**for** *t=1, 2, ...* **do**

   Reject the $t$-th null if $P_t \leq \alpha_t$, where $\alpha_t := \min\{\tau, \widetilde{\alpha}_t\}$, and

   $\widetilde{\alpha}_t := \tau \left( W_0 \gamma_{S^t} + (\alpha - W_0)\gamma_{S^t - \kappa_1^*} + \alpha \sum_{j \geq 2} \gamma_{S^t - \kappa_j^*} \right).$

   Here,

   $\kappa_j = \min\{i \in [t-1] : \sum_{k \leq i} \mathbf{1}\{P_k \leq \alpha_k\} \geq j\}, \quad \kappa_j^* = \sum_{i \leq \kappa_j} \mathbf{1}\{P_i \leq \tau\},$

   $S^t = \sum_{i < t} \mathbf{1}\{P_i \leq \tau\}.$

**end**

---

Here we present the following theorem for error control of D-LORD. Recall the definition of uniformly conservative $p$-values (4); and here we call a function $f_t(R_{1:t-1}, S_{1:t-1}) : \{0,1\}^{2(t-1)} \to [0,1]$ as the "monotonic" function of the past if it is coordinatewise nondecreasing with regard $R_j$, and coordinatewise nonincreasing with regard $S_j$.

**Theorem S-1.** *If the null $p$-values are uniformly conservative, and suppose we choose $\tau_j \geq \alpha_j$ for each $j \in \mathbb{N}$, where $\alpha_j$ is the testing level for $j$-th hypothesis, then we have:*

*(a) any algorithm with $\widehat{\mathrm{FDP}}_{\mathrm{D\text{-}LORD}}(t) \leq \alpha$ for all $t \in \mathbb{N}$ also enjoys $\mathrm{mFDR}(t) \leq \alpha$ for all $t \in \mathbb{N}$. Further, if the null $p$-values are independent of each other and of the non-nulls, and for all $t$, $\alpha_t$ and $1 - \tau_t$ are both monotonic functions of the past, then we additionally have:*

*(b) any algorithm with $\widehat{\mathrm{FDP}}_{\mathrm{D\text{-}LORD}}(t) \leq \alpha$ for all $t \in \mathbb{N}$ also enjoys $\mathrm{FDR}(t) \leq \alpha$ for all $t \in \mathbb{N}$.*
*As an immediate corollary, D-LORD\* (Algorithm S-1) enjoys both mFDR and FDR control.*

The proof of Theorem S-1 is presented in Section S-7.

## S-2   D-StBH: Storey-BH with discarding

The discarding rule can also be applied to offline settings. Here we present the D-StBH, i.e. the discarding version of an adaptive offline FDR control method — Storey-BH [3, 4]. Just as SAFFRON is an online analog of Storey-BH, ADDIS may be regarded as an online analog of D-StBH.

Now we present the specific approach. Denote the number of hypotheses as $n$. Given targeted FDR level $\alpha$, user defined constants $\lambda < \tau \in (0, 1]$, we define

$$\widehat{\mathrm{FDP}}_{\mathrm{D\text{-}StBH}}(s) := \frac{n \cdot s \cdot \widehat{\pi_0}}{(\sum_j \mathbf{1}\{P_j \leq s\}) \vee 1}, \quad \text{where} \quad \widehat{\pi_0} := \frac{1 + \sum_{i=1}^n \mathbf{1}\{\lambda < P_i \leq \tau\}}{n(\tau - \lambda)}. \tag{S-3}$$

D-StBH then calculates $\widehat{s} := \max\{s : s \leq \tau, \widehat{\mathrm{FDP}}_{\mathrm{D\text{-}StBH}}(s) \leq \alpha\}$, and reject the set $\{i : P_i \leq \widehat{s}\}$. With many conservative nulls, we claim D-StBH would be more powerful than Storey-BH, since $\widehat{\pi_0}$ serves as a tighter estimator for the true $\pi_0 := |\mathcal{H}_0|/n$ in terms of expectation. As always, we present the error control of the new method under some reasonable assumptions, and the simulations demonstrating its power advantage in Section 4.

**Theorem S-2.** *If $p$-values are independent with each other and the nulls are uniformly conservative as defined in* (2)*, then* D-StBH *controls* FDR *at level $\alpha$.*

Theorem S-2 is proved in Section S-8 .

## S-3   Asynchronous setting

Here we formalize the asynchronous setting. An asynchronous testing process consists of tests that start and finish at random times. Without loss of generality, one can take the starting times of each tests as $1, 2, \ldots$, and refer them as $H_1, H_2, \ldots$, and take the finish time of each tests as $E_1, E_2, \ldots$ accordingly (let $E_t = j$, if $j \leq E_t < j + 1$). Notice that $E_t$ may be bigger than $t$. One has to decide the testing level for $H_t$ at its starting time, with only information of tests that finished before time $t$. It is worth mentioning that this framework is a generalization of the classical online FDR setting, since it reduces to the classical setting when $E_t = t$ for all $t$. We refer readers to [5] for more detailed definition and discussion.

In the following of the section, we present the modified ADDIS algorithm under asynchronous setting, which we will refer as ADDIS$_{\mathrm{async}}$. We derive the new method respectively from the following two empirical estimators for the oracle metric FDP\* for true FDP, which is

$$\widehat{\mathrm{FDP}}_{\mathrm{ADDIS}_{\mathrm{async}}}(t) := \frac{\sum_{j \leq t} \frac{\alpha_j}{(\tau_j - \lambda_j)} \left(\mathbf{1}\{\lambda_j < P_j \leq \tau_j, E_j < t\} + \mathbf{1}\{E_j \geq t\}\right)}{(\sum_{j \leq t} \mathbf{1}\{P_j \leq \alpha_j, E_j < t\}) \vee 1}. \tag{S-4}$$

As before, $\{\tau_j\}_{j=1}^\infty, \{\lambda_j\}_{j=1}^\infty, \{\alpha_j\}_{j=1}^\infty$ are some user defined sequences, where each terms is in range $[0, 1]$. We use $P_t$ to refer the $p$-value that results from the test started at time $t$, which is not known at time $t$, but only at time $E_t$ (unless they are identical). Similarly, $S_t, C_t, R_t$ are defined in the same way as Section 2, to indicate whether the hypothesis started at time $t$ is selected, candidate of

rejection, or rejected, respectively. Like $P_t$, they are also not known before time $E_t$. Additionally, denote $\mathcal{R}_t = \{i : E_i = t, R_i = 1\}$, $\mathcal{C}_t = \{i : E_i = t, C_i = 1\}$ and $\mathcal{S}_t = \{i : E_i = t, S_i = 1\}$. Correspondingly, denote $\mathcal{R}_{1:t} = \{\mathcal{R}_1, \ldots, \mathcal{R}_t\}$, $\mathcal{C}_{1:t} = \{\mathcal{C}_1, \ldots, \mathcal{C}_t\}$ and $\mathcal{S}_{1:t} = \{\mathcal{S}_1, \ldots, \mathcal{S}_t\}$. As always, we refer the online FDR algorithm as $\mathrm{ADDIS_{async}}$ if it updates $\alpha_t$ to maintain the invariant $\widehat{\mathrm{FDP}}_{\mathrm{ADDIS_{async}}}(t) \le \alpha$ for all $t \in \mathbb{N}$.

Now we present explicit instance for $\mathrm{ADDIS_{async}}$ algorithm for fixed $\tau$ and $\lambda$.

---

**Algorithm S-2:** The $\mathrm{ADDIS^*_{async}}$ algorithm

---

**Input:** FDR level $\alpha$, discarding threshold $\tau \in (0, 1]$, candidate threshold $\lambda \in [0, \tau)$, sequence $\{\gamma_j\}_{j=1}^\infty$ which is nonnegative, nonincreasing and sums to one, initial wealth $W_0 \le \alpha$.

**for** $t = 1, 2, \ldots$ **do**

Start $t$-th test with level $\alpha_t := \min\{\lambda, \widetilde{\alpha}_t\}$,

where $\widetilde{\alpha}_t := (\tau - \lambda)\left(W_0 \gamma_{S^t - C_0^+} + (\alpha - W_0)\gamma_{S^t - \kappa_1^* - C_1^+} + \alpha \sum_{j \ge 2} \gamma_{S^t - \kappa_j^* - C_j^+}\right)$.

Here, $S^t = \sum_{i < t}(\mathbf{1}\{P_i \le \tau, E_i < t\} + \mathbf{1}\{E_i \ge t\})$,

$\qquad C_j^+ = \sum_{i < t} \mathbf{1}\{P_i \le \lambda, \; \kappa_j + 1 \le E_i < t\}$,

$\qquad \kappa_j = \min\{i \in [t-1] : \sum_{k \le t} \mathbf{1}\{P_k \le \alpha_k, E_k \le i\} \ge j\}$,

$\qquad \kappa_j^* = \sum_{i < t} \mathbf{1}\{P_i \le \tau, E_i \le \kappa_j\}$.

**end**

---

As always, we present the error control for the $\mathrm{ADDIS_{async}}$, by proving theorem as the following. Firstly, we clarify the following terms.

Here, we say $P_i$ is uniformly conservative, if it satisfy the uniformly conservative condition defined in (4), with specified filtration $\mathcal{F}^{E_t - 1}$, where $\mathcal{F}^{t-1} = \sigma\{\mathcal{R}_{1:t-1}, \mathcal{C}_{1:t-1}, \mathcal{S}_{1:t-1}\}$. We insist that the thresholds $\tau_j, \lambda_j$ and $\alpha_j$ in $\mathrm{ADDIS_{async}}$ are mappings from $(\mathcal{R}_{1:j-1}, \mathcal{C}_{1:j-1}, \mathcal{S}_{1:j-1})$ to $[0, 1]$ for each $j \in \mathbb{N}$. Here, we say $f_t$ is a monotonic function of the past, if it is nondecreasing in $|\mathcal{R}_j|$ and $|\mathcal{C}_j|$, while nonincreasing in $|\mathcal{S}_j|$.

**Theorem S-3.** *If the null p-values are uniformly conservative, suppose we choose $\tau_j > \lambda_j \ge \alpha_j$ for each $j \in \mathbb{N}$. Then we have:*

*(a) any algorithm with $\widehat{\mathrm{FDP}}_{\mathrm{ADDIS_{async}}}(t) \le \alpha$ for all $t \in \mathbb{N}$ enjoys $\mathrm{mFDR}(t) \le \alpha$ for all $t \in \mathbb{N}$. Next assume that the null p-values are independent of each other and of the non-nulls, and each p-value $P_t$ is independent of its decision time given $\mathcal{F}^{E_t - 1}$. If $\alpha_t, \lambda_t, 1 - \tau_t$ are all designed to be monotonic functions of the past for all $t \in \mathbb{N}$, then we additionally have:*

*(b) any algorithm with $\widehat{\mathrm{FDP}}_{\mathrm{ADDIS_{async}}}(t) \le \alpha$ for all $t \in \mathbb{N}$ enjoys $\mathrm{FDR}(t) \le \alpha$ for all $t \in \mathbb{N}$. As an immediate corollary, $\mathrm{ADDIS^*_{async}}$ (Algorithm S-2) have both mFDR and FDR control.*

Theorem S-3 is proved using Lemma S-4, which is a modified version of Lemma S-1 in Section 2. The proof is presented in Section S-9.

## S-4  Proof of Lemma 1

Let $F_j$ denote the CDF of null p-value $P_j$, for fixed $b \in (0, 1)$, let $h_j(a) = bF_j(a) - F_j(ab)$. Since $F_j$ is differentiable, let $f_j$ denote its density function, and notice that $f_j$ is monotonically increasing by the fact that $F_j$ is convex. Then we have that the derivative of $h_j$ is

$$h'_j(a) = \tau_j f_j(a) - b f_j(ab) \ge 0.$$

Therefore, $h_j$ is increasing with $a$, which implies $h_j(a) \le h_j(1)$. With simple rearrangement, we have

$$\frac{\Pr\{ab < P_j \le b\}}{b(1 - a)} \le \frac{\Pr\{P_j > a\}}{(1 - a)}$$

as claimed.

## S-5   Proof of Theorem 1

Part (a) of Theorem 1 is proved using the the law of iterated expectations and the property of uniformly conservative null $p$-values as stated in (4). Specifically, taking iterated expectation by conditioning on $\{\mathcal{F}^{j-1}, S_j\}$ respectively for each $j \in \mathcal{H}_0$, we have

$$
\begin{aligned}
\mathbb{E}\left[|\mathcal{H}_0 \cap \mathcal{R}(t)|\right] &= \sum_{j \leq t, j \in \mathcal{H}_0} \mathbb{E}\left[\mathbf{1}\{P_j \leq \alpha_j\}\right] \\
&= \sum_{j \leq t, j \in \mathcal{H}_0} \mathbb{E}\left[\mathbb{E}\left[\mathbf{1}\{P_j \leq \alpha_j\} \mid S_j, \mathcal{F}^{j-1}\right]\right] \\
&\stackrel{(i)}{=} \sum_{j \leq t, j \in \mathcal{H}_0} \mathbb{E}\left[\mathbb{E}\left[\mathbf{1}\{\frac{P_j}{\tau_j} \leq \frac{\alpha_j}{\tau_j}\} \ \Big| \ S_j = 1, \mathcal{F}^{j-1}\right] \Pr\{S_j = 1 \mid \mathcal{F}^{j-1}\}\right] \\
&= \sum_{j \leq t, j \in \mathcal{H}_0} \mathbb{E}\left[\mathbb{E}\left[\mathbf{1}\{\frac{P_j}{\tau_j} \leq \frac{\alpha_j}{\tau_j}\} \ \Big| \ P_j \leq \tau_j, \mathcal{F}^{j-1}\right] \Pr\{S_j = 1 \mid \mathcal{F}^{j-1}\}\right],
\end{aligned}
\tag{S-5}
$$

where (i) is true since $\alpha_j < \tau_j$, therefore $P_j \leq \alpha_j$ implies $S_j = 1$. Then, using the property of the uniformly conservative null $p$-values stated in (4), we have

$$
\begin{aligned}
&\sum_{j \leq t, j \in \mathcal{H}_0} \mathbb{E}\left[\mathbb{E}\left[\mathbf{1}\{\frac{P_j}{\tau_j} \leq \frac{\alpha_j}{\tau_j}\} \ \Big| \ P_j \leq \tau_j, \mathcal{F}^{j-1}\right] \Pr\{S_j = 1 \mid \mathcal{F}^{j-1}\}\right] \\
&\leq \sum_{j \leq t, j \in \mathcal{H}_0} \mathbb{E}\left[\frac{\alpha_j}{\tau_j} \Pr\{S_j = 1 \mid \mathcal{F}^{j-1}\}\right] \\
&\leq \sum_{j \leq t, j \in \mathcal{H}_0} \mathbb{E}\left[\frac{\alpha_j}{\tau_j} \mathbb{E}\left[\frac{\mathbf{1}\{\theta_j \tau_j < P_j\}}{(1 - \theta_j)} \ \Big| \ P_j \leq \tau_j, \mathcal{F}^{j-1}\right] \Pr\{S_j = 1 \mid \mathcal{F}^{j-1}\}\right],
\end{aligned}
\tag{S-6}
$$

where $\theta_j \equiv \lambda_j / \tau_j$. Next, using the fact that $\alpha_j, \lambda_j$ and $\tau_j$ are measurable with regard $\mathcal{F}^{j-1}$ for all $j \in \mathbb{N}$, the RHS of (S-6) equals

$$
\begin{aligned}
&\sum_{j \leq t, j \in \mathcal{H}_0} \mathbb{E}\left[\mathbb{E}\left[\frac{\alpha_j}{\tau_j} \frac{\mathbf{1}\{\theta_j \tau_j < P_j\}}{(1 - \theta_j)} \ \Big| \ P_j \leq \tau_j, \mathcal{F}^{j-1}\right] \Pr\{S_j = 1 \mid \mathcal{F}^{j-1}\}\right] \\
&= \sum_{j \leq t, j \in \mathcal{H}_0} \mathbb{E}\left[\mathbb{E}\left[\alpha_j \frac{\mathbf{1}\{\lambda_j < P_j \leq \tau_j\}}{(\tau_j - \lambda_j)} \ \Big| \ S_j = 1, \mathcal{F}^{j-1}\right] \Pr\{S_j = 1 \mid \mathcal{F}^{j-1}\}\right] \\
&= \sum_{j \leq t, j \in \mathcal{H}_0} \mathbb{E}\left[\mathbb{E}\left[\alpha_j \frac{\mathbf{1}\{\lambda_j < P_j \leq \tau_j\}}{(\tau_j - \lambda_j)} \ \Big| \ S_j, \mathcal{F}^{j-1}\right]\right] \\
&\stackrel{(ii)}{=} \sum_{j \leq t, j \in \mathcal{H}_0} \mathbb{E}\left[\alpha_j \frac{\mathbf{1}\{\lambda_j < P_j \leq \tau_j\}}{(\tau_j - \lambda_j)}\right] \stackrel{(iii)}{=} \mathbb{E}\left[\sum_{j \leq t, j \in \mathcal{H}_0} \alpha_j \frac{\mathbf{1}\{\lambda_j < P_j \leq \tau_j\}}{(\tau_j - \lambda_j)}\right],
\end{aligned}
\tag{S-7}
$$

where (ii) is again obtained using law of the iterated expectations; and (iii) is obtained using the linearity of expectation. Therefore, combine the results above, we have

$$
\mathbb{E}\left[|\mathcal{H}_0 \cap R(t)|\right] \leq \mathbb{E}\left[\sum_{j \leq t, j \in \mathcal{H}_0} \alpha_j \frac{\mathbf{1}\{\lambda_j < P_j \leq \tau_j\}}{(\tau_j - \lambda_j)}\right].
\tag{S-8}
$$

Furthermore, since

$$
\widehat{\mathrm{FDP}}_{\mathrm{ADDIS}}(t) \leq \alpha \Rightarrow \sum_{j \leq t} \alpha_j \frac{\mathbf{1}\{\lambda_j < P_j \leq \tau_j\}}{(\tau_j - \lambda_j)} \leq \alpha(|R(t)| \vee 1),
$$

take expectation on each side and use (S-8), we have $\mathrm{mFDR}(t) \leq \alpha$ as claimed.

Next, in order to prove part (b), we need Lemma S-1 in the following, which is a modified version of "reverse super-uniformity lemma" in [6]. Recall the definition of "monotonic (neg-montonic) function of the past" in 2.2, we present Lemma S-1 as follows.

**Lemma S-1.** *Assume that the p-values $P_1, P_2, \ldots$ are independent and let $g : \{0,1\}^T \to \mathbb{R}$ be any coordinatewise nondecreasing function, and assume $\alpha_t$, $\lambda_t$ and $1 - \tau_t$ are all monotonic function of the past as defined in 2.2, while satisfying the constraints $\alpha_t \leq \lambda_t < \tau_t$ for all t. Then, for any index $t \leq T$ such that $t \in \mathcal{H}_0$, we have:*

$$\mathbb{E}\left[\frac{\alpha_t \mathbf{1}\{\lambda_t < P_t \leq \tau_t\}}{(\tau_t - \lambda_t)(g(R_{1:T}) \vee 1)} \,\bigg|\, \mathcal{F}^{t-1}, S_t = 1\right] \geq \mathbb{E}\left[\frac{\alpha_t}{\tau_t(g(R_{1:T}) \vee 1)} \,\bigg|\, \mathcal{F}^{t-1}, S_t = 1\right]$$

$$\geq \mathbb{E}\left[\frac{\mathbf{1}\{P_t \leq \alpha_t\}}{g(R_{1:T}) \vee 1} \,\bigg|\, \mathcal{F}^{t-1}, S_t = 1\right].$$

The proof of Lemma S-1 is deferred in Section S-5.1.

Now, taking iterated expectations similarly as in the proof of part (a), we obtain the following:

$$\text{FDR}(t) = \mathbb{E}\left[\text{FDP}(t)\right] = \mathbb{E}\left[\frac{|\mathcal{H}_0 \cap R(t)|}{|R(t)| \vee 1}\right] = \sum_{j \leq t, j \in \mathcal{H}_0} \mathbb{E}\left[\frac{\mathbf{1}\{P_j \leq \alpha_j\}}{|R(t)| \vee 1}\right]$$

$$= \sum_{j \leq t, j \in \mathcal{H}_0} \mathbb{E}\left[\mathbb{E}\left[\frac{\mathbf{1}\{P_j \leq \alpha_j\}}{|R(t)| \vee 1} \,\bigg|\, S_j, \mathcal{F}^{j-1}\right]\right]$$

$$= \sum_{j \leq t, j \in \mathcal{H}_0} \mathbb{E}\left[\mathbb{E}\left[\frac{\mathbf{1}\{P_j \leq \alpha_j\}}{|R(t)| \vee 1} \,\bigg|\, S_j = 1, \mathcal{F}^{j-1}\right] \Pr\{S_j = 1 \mid \mathcal{F}^{j-1}\}\right]$$

(S-9)

Under the independence and monotonicity assumptions of part (b), and notice that $|R(t)| = \sum_{i=1}^{t} R_i$ is a coordinatewise nondecreasing function with regard $R_{1:t}$, we use Lemma S-1 to obtain the following:

$$\sum_{j \leq t, j \in \mathcal{H}_0} \mathbb{E}\left[\mathbb{E}\left[\frac{\mathbf{1}\{P_j \leq \alpha_j\}}{|R(t)| \vee 1} \,\bigg|\, S_j = 1, \mathcal{F}^{j-1}\right] \Pr\{S_j = 1 \mid \mathcal{F}^{j-1}\}\right]$$

$$\leq \sum_{j \leq t, j \in \mathcal{H}_0} \mathbb{E}\left[\mathbb{E}\left[\frac{\alpha_j}{\tau_j(|R(t)| \vee 1)} \,\bigg|\, \mathcal{S}_j = 1, \mathcal{F}^{j-1}\right] \Pr\{\mathcal{S}_j = 1 \mid \mathcal{F}^{j-1}\}\right]$$

$$\leq \sum_{j \leq t, j \in \mathcal{H}_0} \mathbb{E}\left[\mathbb{E}\left[\frac{\alpha_j \mathbf{1}\{\lambda_j < P_j \leq \tau_j\}}{(\tau_j - \lambda_j)(|R(t)| \vee 1)} \,\bigg|\, S_j = 1, \mathcal{F}^{j-1}\right] \Pr\{S_j = 1 \mid \mathcal{F}^{j-1}\}\right] \quad \text{(S-10)}$$

Again using the law of iterated expectation and the linearity of expectation, we have the RHS of (S-10) equals

$$\sum_{j \leq t, j \in \mathcal{H}_0} \mathbb{E}\left[\mathbb{E}\left[\frac{\alpha_j \mathbf{1}\{\lambda_j < P_j \leq \tau_j\}}{(\tau_j - \lambda_j)(|R(t)| \vee 1)} \,\bigg|\, S_j, \mathcal{F}^{j-1}\right]\right]$$

$$= \sum_{j \leq t, j \in \mathcal{H}_0} \mathbb{E}\left[\frac{\alpha_j \mathbf{1}\{\lambda_j < P_j \leq \tau_j\}}{(\tau_j - \lambda_j)(|R(t)| \vee 1)}\right]$$

$$= \mathbb{E}\left[\frac{1}{|R(t)| \vee 1} \sum_{j \leq t, j \in \mathcal{H}_0} \alpha_j \frac{\mathbf{1}\{\lambda_j < P_j \leq \tau_j\}}{(\tau_j - \lambda_j)}\right], \quad \text{(S-11)}$$

which is no larger than $\mathbb{E}\left[\widehat{\text{FDP}}_{\text{ADDIS}}(t)\right] \leq \alpha$ by the definition of $\widehat{\text{FDP}}_{\text{ADDIS}}(t)$. Therefore, combine (S-9), (S-10) and (S-11), we have $\text{FDR}(t) \leq \alpha$ as claimed.

Finally, we justify for the corollary that $\text{ADDIS}^*$ have mFDR and FDR control. Firstly, from Algorithm 1, we know that $\text{ADDIS}^*$ makes sure $\tau > \lambda \geq \alpha_j$ for all $j$, and constant $\lambda$ and $1 - \tau$ is obviously monotonic function of the past, while $\alpha_t$ being a monotonic function of the past for all $t$ is verified in Section S-5.2. Then, from the definition of sequence $\{\gamma_j\}_{j=0}^{\infty}$, after simple rearrangement, we have $\widehat{\text{FDP}}_{\text{ADDIS}}(t) \leq \alpha$ holds true. Therefore, $\text{ADDIS}^*$ satisfy all the requirements in the theorem, thus having error control under corresponding assumptions of p-values.

### S-5.1 Proof of Lemma S-1

We use a technique of constructing a hallucinated vector, similar to [6], to prove Lemma S-1. Specifically, to prove the first part of the inequality, first fix the time $t$, and then construct a hallucinated vector $\widetilde{P}$, such that for each $i \in \mathbb{N}$,

$$\widetilde{P}_i = \tau_i \cdot \mathbf{1}\{i = t\} + P_i \cdot \mathbf{1}\{i \neq t\}. \tag{S-12}$$

Denote the corresponding hallucinated testing levels, candidate levels and selected levels resulting from $\{\widetilde{P}_i\}$ as $\{\widetilde{\alpha}_i\}$, $\{\widetilde{\lambda}_i\}$ and $\{\widetilde{\tau}_j\}$ respectively. Similarly, we define the corresponding hallucinated indicator variables as

$$\widetilde{S}_i = \mathbf{1}\{\widetilde{P}_i \leq \widetilde{\tau}_i\}, \quad \widetilde{C}_i = \mathbf{1}\{\widetilde{P}_i \leq \widetilde{\lambda}_i\}, \quad \widetilde{R}_i = \mathbf{1}\{\widetilde{P}_i \leq \widetilde{\alpha}_i\}.$$

Given $\lambda_t < P_t \leq \tau_t$, we have $\widetilde{S}_t = S_t = 1$, $\widetilde{R}_t = R_t = 0$, $\widetilde{C}_t = C_t = 0$. Therefore, $R_{1:T} = \widetilde{R}_{1:T}$, and particularly $\widetilde{R}_{1:T}$ is independent of $P_t$. These facts lead to:

$$\mathbb{E}\left[\frac{\alpha_t \mathbf{1}\{\lambda_t < P_t \leq \tau_t\}}{(\tau_t - \lambda_t)(g(R_{1:T}) \vee 1)}\,\middle|\, S_t = 1, \mathcal{F}^{t-1}\right]$$

$$= \mathbb{E}\left[\frac{\alpha_t \mathbf{1}\{\lambda_t < P_t \leq \tau_t\}}{(\tau_t - \lambda_t)(g(\widetilde{R}_{1:T}) \vee 1)}\,\middle|\, S_t = 1, \mathcal{F}^{t-1}\right]$$

$$\overset{(i)}{=} \mathbb{E}\left[\frac{\alpha_t}{\tau_t(g(\widetilde{R}_{1:T}) \vee 1)}\,\middle|\, S_t = 1, \mathcal{F}^{t-1}\right]\mathbb{E}\left[\frac{\mathbf{1}\{\lambda_t < P_t \leq \tau_t\}}{(1 - \lambda_t/\tau_t)}\,\middle|\, S_t = 1, \mathcal{F}^{t-1}\right]$$

$$\overset{(ii)}{\geq} \mathbb{E}\left[\frac{\alpha_t}{\tau_t(g(\widetilde{R}_{1:T}) \vee 1)}\,\middle|\, S_t = 1, \mathcal{F}^{t-1}\right],$$

where (i) is obtained from the fact that $\widetilde{R}_{1:T}$ is independent of $P_t$, and that $\lambda_t, \tau_t, f_t$ are measurable with regard $\mathcal{F}^{t-1}$; (ii) is obtained using the property of uniformly conservative null $p$-values stated in (4).

Under the construction of hallucinated variables, if $S_t = 1$, then $\widetilde{R}_i \leq R_i$ for all $i \in \mathbb{N}$. This statement follows by the monotonicity of $\{\alpha_i\}$ and $\{\lambda_i\}$, and the neg-monotonicity of $\{\tau_i\}$. Notice that for all $i < t$, we have $\widetilde{S}_i = S_i$, $\widetilde{R}_i = R_i$, $\widetilde{C}_i = C_i$. Therefore, we may infer that $\alpha_i = \widetilde{\alpha}_i$, $\lambda_i = \widetilde{\lambda}_i$ and $\tau_i = \widetilde{\tau}_i$ for all $i \leq t$. Since $\widetilde{S}_t = S_t = 1$, $\widetilde{R}_t = \widetilde{C}_t = 0$, that is $\widetilde{S}_t = S_t$, $\widetilde{C}_t \leq C_t$, $\widetilde{R}_t \leq R_t$. Therefore we have that $\widetilde{\alpha}_{t+1} \leq \alpha_{t+1}$, $\widetilde{\lambda}_{t+1} \leq \lambda_{t+1}$ and $\widetilde{\tau}_{t+1} \geq \tau_{t+1}$, which lead to $\widetilde{R}_{t+1} \leq R_{t+1}$, $\widetilde{C}_{t+1} \leq C_{t+1}$ and $\widetilde{S}_{t+1} \geq S_{t+1}$ and so on. Recursively, we deduce $\widetilde{R}_{t+1} \leq R_{t+1}$ for all $i > t$. Since $g$ is a coordinatewise increasing function, we have

$$\mathbb{E}\left[\frac{\alpha_t}{\tau_t(g(\widetilde{R}_{1:T}) \vee 1)}\,\middle|\, S_t = 1, \mathcal{F}^{t-1}\right] \leq \mathbb{E}\left[\frac{\alpha_t}{\tau_t(g(R_{1:T}) \vee 1)}\,\middle|\, S_t = 1, \mathcal{F}^{t-1}\right] \tag{S-13}$$

Hence, we proved the first part of inequality in Lemma S-1.

To prove the second part of the inequality, alternatively, for all $t \in \mathbb{N}$, we let $\widetilde{P}_i = P_i \cdot \mathbf{1}\{i \neq t\}$, and define $\widetilde{\alpha}_i, \widetilde{\lambda}_i, \widetilde{\tau}_i$ and $\widetilde{S}_i, \widetilde{C}_i, \widetilde{R}_i$ in same way as before.

On the other hand, given $P_t \leq \alpha_t$, we have $\widetilde{S}_t = S_t = \widetilde{R}_t = R_t = \widetilde{C}_t = C_t = 1$. Therefore, $R_{1:T} = \widetilde{R}_{1:T}$, and particularly $\widetilde{R}_{1:T}$ is independent of $P_t$. Again, we have:

$$\mathbb{E}\left[\frac{\mathbf{1}\{P_t \leq \alpha_t\}}{g(R_{1:T}) \vee 1}\,\middle|\, S_t = 1, \mathcal{F}^{t-1}\right] = \mathbb{E}\left[\frac{\mathbf{1}\{P_t \leq \alpha_t\}}{g(\widetilde{R}_{1:T}) \vee 1}\,\middle|\, S_t = 1, \mathcal{F}^{t-1}\right]$$

$$\overset{(i)}{=} \mathbb{E}\left[\frac{\alpha_t}{\tau_t(g(\widetilde{R}_{1:T}) \vee 1)}\,\middle|\, S_t = 1, \mathcal{F}^{t-1}\right]\mathbb{E}\left[\frac{\mathbf{1}\{P_t \leq \alpha_t\}}{\alpha_t/\tau_t}\,\middle|\, S_t = 1, \mathcal{F}^{t-1}\right]$$

$$\overset{(ii)}{\leq} \mathbb{E}\left[\frac{\alpha_t}{\tau_t(g(\widetilde{R}_{1:T}) \vee 1)}\,\middle|\, S_t = 1, \mathcal{F}^{t-1}\right] \overset{(iii)}{\leq} \mathbb{E}\left[\frac{\alpha_t}{\tau_t(g(R_{1:T}) \vee 1)}\,\middle|\, S_t = 1, \mathcal{F}^{t-1}\right],$$

where (i) is obtained from the fact that $\widetilde{R}_{1:T}$ is independent of $P_t$, and that $\lambda_t, \tau_t, f_t$ are measurable with regard $\mathcal{F}^{t-1}$; and (ii) is true due to the property of uniformly conservative $p$-values stated in (4) ; and finally, (iii) is true from the similar logic in the proof of first part.

These concludes the proof of the second part of inequality in Lemma S-1.

### S-5.2 Verify $\alpha_t$ in ADDIS* is a monotonic function of the past

In applying Theorem 1 to prove that ADDIS* controls the FDR, it is assumed that ADDIS* is a monotonic rule, meaning that $\alpha_t$ is a monotonic function of the past as defined in 2.2. Here we justify for this claim. In ADDIS*, we assume $\lambda$ and $\tau$ is constant, however the same arguments can be applied if they change at every step, but are predictable as stated in Section 2 of the main paper.

We will prove this argument by proving that $\alpha_t$ in ADDIS* satisfy some equivalent argument of monotonicity defined in 2.2. Consider some $(R_{1:t-1}, C_{1:t-1}, S_{1:t-1})$ and $(\widetilde{R}_{1:t-1}, \widetilde{C}_{1:t-1}, \widetilde{S}_{t-1})$ for a fixed $t$. We will accordingly denote all relevant variables in the ADDIS* alogorithm which result in $(R_{1:t-1}, C_{1:t-1}, S_{1:t-1})$ and $(\widetilde{R}_{1:t-1}, \widetilde{C}_{1:t-1}, \widetilde{S}_{1:t-1})$, e.g. $\alpha_t$ and $\widetilde{\alpha}_t$, respectively. We say $(\widetilde{R}_{1:t-1}, \widetilde{C}_{1:t-1}, \widetilde{S}_{1:t-1}) \succeq (R_{1:t-1}, C_{1:t-1}, S_{t-1})$ if and only if, for each $i \leq t-1$, one of the following holds:

(1) $R_i = \widetilde{R}_i$, $C_i = \widetilde{C}_i$, and $R_i = \widetilde{R}_i$;

(2) $R_i = 0$, $C_i = 0$, $S_i = 1$, and $\widetilde{R}_i = 0$, $\widetilde{C}_i = 1$, $\widetilde{S}_i = 1$;

(3) $R_i = 0$, $C_i = 0$, $S_i = 1$, and $\widetilde{R}_i = 1$, $\widetilde{C}_i = 1$, $\widetilde{S}_i = 1$;

(4) $R_i = 0$, $C_i = 1$, $S_i = 1$, and $\widetilde{R}_i = 1$, $\widetilde{C}_i = 1$, $\widetilde{S}_i = 1$.

(5) $R_i = 0$, $C_i = 0$, $S_i = 1$, and $\widetilde{R}_i = 0$, $\widetilde{C}_i = 0$, $\widetilde{S}_i = 0$;

Taking into account the possible relations between indicators for rejection, candidacy and tester, one may notice the fact that $S_i \geq C_i \geq R_i$ for each $i$. Then the monotonicity defined in 2.2 of a function $\alpha_t$ is equivalent to the statement that $(\widetilde{R}_{1:t-1}, \widetilde{C}_{1:t-1}, \widetilde{S}_{1:t-1}) \succeq (R_{1:t-1}, C_{1:t-1}, S_{1:t-1})$ implies $\widetilde{\alpha}_t \geq \alpha_t$. Therefore, we will instead prove that this equivalent statement holds for $\alpha_t$ in ADDIS* for each $t \in \mathbb{N}$. Specifically, recall the forms of $\alpha_t$ in ADDIS*:

$$\alpha_t := \min\{\lambda, \widehat{\alpha}_t\},$$

$$\text{where } \widehat{\alpha}_t := (\tau - \lambda)\left(W_0 \gamma_{S^t - C_{0+}} + (\alpha - W_0)\gamma_{S^t - \kappa_1^* - C_{1+}} + \alpha \sum_{j \geq 2} \gamma_{S^t - \kappa_j^* - C_{j+}}\right). \tag{S-14}$$

We would like to prove that, given $(\widetilde{R}_{1:t-1}, \widetilde{C}_{1:t-1}, \widetilde{S}_{1:t-1}) \succeq (R_{1:t-1}, C_{1:t-1}, S_{1:t-1})$, we have $\widetilde{\alpha}_t \geq \alpha_t$. First, notice that in (S-14), the index $S^t - \kappa_j^* - C_{j+}$ is the number of non-candidate testers (i.e. $\{i : S_i = 1, C_i = 0\}$) between the $j$-th rejection before time $t$ and time $t$. Provided with $(\widetilde{R}_{1:t-1}, \widetilde{C}_{1:t-1}, \widetilde{S}_{1:t-1}) \succeq (R_{1:t-1}, C_{1:t-1}, S_{1:t-1})$, we must have that $(R_{1:t-1}, C_{1:t-1}, S_{1:t-1})$ never contains less non-candidate testers or more rejections compared to $(\widetilde{R}_{1:t-1}, \widetilde{C}_{1:t-1}, \widetilde{S}_{1:t-1})$, from the definition of $(\widetilde{R}_{1:t-1}, \widetilde{C}_{1:t-1}, \widetilde{S}_{1:t-1}) \succeq (R_{1:t-1}, C_{1:t-1}, S_{1:t-1})$ above. Additionally, notice that the sequence $\{\gamma_j\}_{j=0}^{\infty}$ is nonincreasing and nonnegative, and $W_0$, $\alpha - W_0$ and $\tau - \lambda$ in (S-14) are strictly positive by construction. Therefore, the sum of the terms $\gamma_{S^t - \kappa_j^* - C_{j+}}$ contributing to $\alpha_t$ is at most as great as the the sum of the terms $\gamma_{\widetilde{S}^t - \widetilde{\kappa}_j^* - \widetilde{C}_{j+}}$, and the same holds for the terms with $W_0$ and $(\alpha - W_0)$. Consequently, we have $\widetilde{\alpha}_t \geq \alpha_t$. Therefore, ADDIS* is a monotonic rule as claimed.

## S-6 Proof of Theorem 2

Using a similar technique to [5], we prove this theorem by constructing a process which behaves similarly to a submartingale, so that we could obtain a result by mimicking optimal stopping. Specifically, for $t \in \mathbb{N}$, define the process $A(t)$ as:

$$A(t) := \sum_{i \le t, i \in \mathcal{H}_0} \left( -\mathbf{1}\{P_j \le \alpha_j\} + \frac{\alpha_j}{(\tau_j - \lambda_j)} \mathbf{1}\{\lambda_j \le P_j < \tau_j\} \right),$$

where we take $A(0) = 0$. Denote $R(t)$ as the set of all rejections made by time $t$, and $V(t)$ as the set of false rejections made by time $t$. Then, we bound

$$
\begin{aligned}
A(t) &= \sum_{i \le t, i \in \mathcal{H}_0} \left( -\mathbf{1}\{P_j \le \alpha_j\} + \frac{\alpha_j}{(\tau_j - \lambda_j)} \mathbf{1}\{\lambda_j \le P_j < \tau_j\} \right) \\
&\le -|V(t)| + \sum_{j \le t} \frac{\alpha_j}{(\tau_j - \lambda_j)} \mathbf{1}\{\lambda_j < P_j \le \tau_j\} \\
&= \alpha(|R(t)| \vee 1) - V(t) + \sum_{j \le t} \frac{\alpha_j}{(\tau_j - \lambda_j)} \mathbf{1}\{\lambda_j < P_j \le \tau_j\} - \alpha(|R(t)| \vee 1) \\
&\overset{(i)}{\le} \alpha(|R(t)| \vee 1) - V(t),
\end{aligned}
$$

where (i) is obtained using the fact that $\text{FDP}_{\text{ADDIS}}(t) \le \alpha$ for all $t$. Therefore, if we can prove $A(T_{\text{stop}}) \ge 0$ for any stopping time $T_{\text{stop}}$ with finite expectation, then we instantly obtain $\alpha|R(T_{\text{stop}})| \ge V(T_{\text{stop}})$. Taking expectation on both side, and rearranging the terms, we obtain $\text{mFDR}(T_{\text{stop}}) \le \alpha$ as claimed.

In order to prove $A(T_{\text{stop}}) \ge 0$ for any stopping time $T_{\text{stop}}$ with finite expectation, we need the following lemma, which is proved in Section S-6.1.

**Lemma S-2.** *If* $\min\{\tau_j - \lambda_j\} > \epsilon$ *for some* $\epsilon > 0$*, and* $T$ *is a random variable supported on* $\mathbb{N}$ *with finite expectation, then the random variable*

$$Y := A(T) \equiv \sum_{j \le T, j \in \mathcal{H}_0} \left( \mathbf{1}\{P_j \le \alpha_j\} + \frac{\alpha_j}{(\tau_j - \lambda_j)} \mathbf{1}\{\lambda_j \le P_j < \tau_j\} \right)$$

*also has finite expectation.*

Since $A(T_{\text{stop}} \wedge t) \to A(T_{\text{stop}})$ almost surely as $t \to \infty$, using Lemma S-2 and the dominate convergence theorem, we conclude that

$$\mathbb{E}\left[A(T_{\text{stop}} \wedge t)\right] \to \mathbb{E}\left[A(T_{\text{stop}})\right], \quad \text{as } t \to \infty. \tag{S-15}$$

Additionally notice that

$$\mathbb{E}\left[A(T_{\text{stop}} \wedge t)\right] = \mathbb{E}\left[A(T_{\text{stop}} \wedge t) - A(0)\right] = \mathbb{E}\left[(H \cdot A)(t)\right], \tag{S-16}$$

where

$$(H \cdot A)(t) := \sum_{m=1}^{t} H(m)(A(m) - A(m-1)), \text{ and } H(t) := \mathbf{1}\{T_{\text{stop}} \ge t\}.$$

Since $T_{\text{stop}}$ is a stopping tome, it holds that $\{T_{\text{stop}} \ge t\} = \{T_{\text{stop}} \le t\}^c \in \mathcal{F}^{t-1}$, therefore $H(t+1)$ is measurable with respect to $\mathcal{F}^t$. Taking conditional expectation, we have:

$$\mathbb{E}\left[(H \cdot A)(t+1) \mid \mathcal{F}^t, S_{t+1}\right]$$
$$= \mathbb{E}\left[(H \cdot A)(t) \mid \mathcal{F}^t, S_{t+1}\right] + \mathbb{E}\left[H(t+1)(A(t+1) - A(t)) \mid \mathcal{F}^t, S_{t+1}\right]$$
$$\overset{(i)}{=} \mathbb{E}\left[(H \cdot A)(t) \mid \mathcal{F}^t, S_{t+1}\right]$$
$$\quad + H(t+1)\mathbf{1}\{t+1 \in \mathcal{H}_0\}\mathbb{E}\left[ -\mathbf{1}\{P_{t+1} \le \alpha_{t+1}\} + \frac{\alpha_{t+1}}{(\tau_{t+1} - \lambda_{t+1})} \mathbf{1}\{\lambda_{t+1} < P_{t+1} \le \tau_{t+1}\} \,\middle|\, \mathcal{F}^t, S_{t+1} \right]$$
$$\overset{(ii)}{\ge} \mathbb{E}\left[(H \cdot A)(t) \mid \mathcal{F}^t, S_{t+1}\right] + H(t+1)\mathbf{1}\{t+1 \in \mathcal{H}_0\}(-\alpha_{t+1}/\tau_{t+1} + \alpha_{t+1}/\tau_{t+1})\mathbf{1}\{S_{t+1} = 1\}$$
$$= \mathbb{E}\left[(H \cdot A)(t) \mid \mathcal{F}^t, S_{t+1}\right],$$

where (i) is obtained from the predictability of $H(t+1)$ with respect to $\mathcal{F}^t$, and the definition of $A(t)$; and (ii) is obtained using the uniform conservative property (4) of nulls.

Therefore, additionally applying the law of iterated expectation, we can have that:

$$\mathbb{E}\left[(H \cdot A)(t+1)\right] \geq \mathbb{E}\left[(H \cdot A)(t)\right].$$

Iteratively applying the same argument, we reach the conclusion that, for all $t \in \mathcal{N}$ :

$$\mathbb{E}\left[(H \cdot A)(t)\right] \geq 0. \tag{S-17}$$

Combining with (S-15) and (S-16), we have that, for any stopping time $T_{\text{stop}}$ with finite expectation, $A(T_{\text{stop}}) \geq 0$ , which leads to $\text{mFDR}(T_{\text{stop}}) \leq \alpha$ as we discussed in the beginning.

### S-6.1   Proof of Lemma S-2

We prove this lemma using an equivalent form of Y. Specifically, notice that we can reformulate $Y$ as:

$$Y = \sum_{j=1}^{\infty} \left( \mathbf{1}\{P_j \leq \alpha_j\} + \frac{\alpha_j}{(\tau_j - \lambda_j)} \mathbf{1}\{\lambda_j < P_j \leq \tau_j\} \right) \mathbf{1}\{j \leq T\}.$$

From the condition that $\min\{\tau_j - \lambda_j\} \geq \epsilon$, we have

$$\mathbf{1}\{P_j \leq \alpha_j\} + \frac{\alpha_j}{(\tau_j - \lambda_j)} \mathbf{1}\{\lambda_j < P_j \leq \tau_j\} \leq 1 + \frac{1}{\epsilon} := C \quad \text{for all } j.$$

Thus, we can bound the expectation of $Y$ as:

$$\mathbb{E}\left[Y\right] = \mathbb{E}\left[\sum_{j=1}^{\infty} \left( \mathbf{1}\{P_j \leq \alpha_j\} + \frac{\alpha_j}{(\tau_j - \lambda_j)} \mathbf{1}\{\lambda_j \leq P_j < \tau_j\} \right) \mathbf{1}\{j \leq T\}\right]$$

$$\leq C \sum_{j=1}^{\infty} \Pr\{T \geq j\} = C \, \mathbb{E}\left[T\right] < \infty.$$

Therefore, we conclude that $Y$ has finite expectation as claimed.

## S-7   Proof of Theorem S-1

Similar to the proof of Theorem 1, part (a) of Theorem S-1 is proved using the property of uniformly conservative null $p$-values as stated in (4), and the law of iterated expectation. Specifically, conditioning on $\{\mathcal{F}^{j-1}, S_j\}$ respectively for each $j \in \mathcal{H}_0$, we have

$$\mathbb{E}\left[|\mathcal{H}_0 \cap R(t)|\right] = \sum_{j \leq t, j \in \mathcal{H}_0} \mathbb{E}\left[\mathbf{1}\{P_j \leq \alpha_j\}\right] = \sum_{j \leq t, j \in \mathcal{H}_0} \mathbb{E}\left[\mathbb{E}\left[\mathbf{1}\{P_j \leq \alpha_j\} \mid S_j, \mathcal{F}^{j-1}\right]\right]$$

$$\stackrel{(i)}{=} \sum_{j \leq t, j \in \mathcal{H}_0} \mathbb{E}\left[\mathbb{E}\left[\mathbf{1}\{\frac{P_j}{\tau_j} \leq \frac{\alpha_j}{\tau_j}\} \,\middle|\, P_j \leq \tau_j, \mathcal{F}^{j-1}\right] \Pr\{S_j = 1 \mid \mathcal{F}^{j-1}\}\right]$$

$$\stackrel{(ii)}{\leq} \sum_{j \leq t, j \in \mathcal{H}_0} \mathbb{E}\left[\frac{\alpha_j}{\tau_j} \Pr\{S_j = 1 \mid \mathcal{F}^{j-1}\}\right]$$

$$\stackrel{(iii)}{=} \sum_{j \leq t, j \in \mathcal{H}_0} \mathbb{E}\left[\mathbb{E}\left[\frac{\alpha_j}{\tau_j}\mathbf{1}\{P_j \leq \tau_j\} \,\middle|\, P_j \leq \tau_j, \mathcal{F}^{j-1}\right] \Pr\{S_j = 1 \mid \mathcal{F}^{j-1}\}\right]$$

$$= \sum_{j \leq t, j \in \mathcal{H}_0} \mathbb{E}\left[\mathbb{E}\left[\frac{\alpha_j}{\tau_j}\mathbf{1}\{P_j \leq \tau_j\} \,\middle|\, S_j, \mathcal{F}^{j-1}\right]\right]$$

$$\stackrel{(iv)}{=} \sum_{j \leq t, j \in \mathcal{H}_0} \mathbb{E}\left[\frac{\alpha_j}{\tau_j}\mathbf{1}\{P_j \leq \tau_j\}\right] = \mathbb{E}\left[\sum_{j \leq t, j \in \mathcal{H}_0} \frac{\alpha_j}{\tau_j}\mathbf{1}\{P_j \leq \tau_j\}\right],$$

where (i) is true since $\alpha_j < \tau_j$ for any $j$; (ii) is obtained using the uniformly conservative property of null $p$-values; (iii) is true since $\alpha_j$ and $\tau_j$ are both predictable given $\mathcal{F}^{j-1}$; and (iv) is obtained using the law of iterated expectation. Therefore, we reach the conclusion that

$$\mathbb{E}\left[|\mathcal{H}_0 \cap R(t)|\right] \le \mathbb{E}\left[\sum_{j \le t, j \in \mathcal{H}_0} \alpha_j \frac{\mathbf{1}\{P_j \le \tau_j\}}{\tau_j}\right]. \tag{S-18}$$

Furthermore, since

$$\widehat{\mathrm{FDP}}_{\text{D-LORD}}(t) \le \alpha \Rightarrow \sum_{j \le t} \alpha_j \frac{\mathbf{1}\{P_j \le \tau_j\}}{\tau_j} \le \alpha(|R(t)| \vee 1),$$

take expectation on each side and use (S-18), we obtain $\mathrm{mFDR}(t) \le \alpha$ with simple rearrangement, which concludes the proof of part (a).

Additionally, under the independence and monotonicity assumption of part (b), using Lemma S-1 with simple modification, together with the same trick of taking iterated expectation and repeatedly using the definition of uniformly conservative nulls, we have the following:

$$
\begin{aligned}
\mathrm{FDR}(t) = \mathbb{E}\left[\mathrm{FDP}(t)\right] &= \mathbb{E}\left[\frac{|\mathcal{H}_0 \cap R(t)|}{|R(t) \vee 1|}\right] = \sum_{j \le t, j \in \mathcal{H}_0} \mathbb{E}\left[\frac{\mathbf{1}\{P_j \le \alpha_j\}}{|R(t)| \vee 1}\right] \\
&= \sum_{j \le t, j \in \mathcal{H}_0} \mathbb{E}\left[\mathbb{E}\left[\frac{\mathbf{1}\{P_j \le \alpha_j\}}{|R(t)| \vee 1} \,\bigg|\, S_j, \mathcal{F}^{j-1}\right]\right] \\
&= \sum_{j \le t, j \in \mathcal{H}_0} \mathbb{E}\left[\mathbb{E}\left[\frac{\mathbf{1}\{P_j \le \alpha_j\}}{|R(t)| \vee 1} \,\bigg|\, S_j = 1, \mathcal{F}^{j-1}\right] \Pr\{S_j = 1 \mid \mathcal{F}^{j-1}\}\right] \\
&\le \sum_{j \le t, j \in \mathcal{H}_0} \mathbb{E}\left[\mathbb{E}\left[\frac{\alpha_j}{\tau_j(|R(t)| \vee 1)} \,\bigg|\, S_j = 1, \mathcal{F}^{j-1}\right] \Pr\{S_j = 1 \mid \mathcal{F}^{j-1}\}\right] \\
&= \sum_{j \le t, j \in \mathcal{H}_0} \mathbb{E}\left[\mathbb{E}\left[\frac{\alpha_j}{\tau_j(|R(t)| \vee 1)} \,\bigg|\, S_j, \mathcal{F}^{j-1}\right]\right] \\
&= \sum_{j \le t, j \in \mathcal{H}_0} \mathbb{E}\left[\frac{\alpha_j}{\tau_j(|R(t)| \vee 1)}\right] \\
&\le \mathbb{E}\left[\widehat{\mathrm{FDP}}_{\text{D-LORD}}(t)\right] \le \alpha.
\end{aligned}
\tag{S-19}
$$

This concludes the proof of statement (b).

## S-8  Proof of Theorem S-2

We will prove this theorem using the trick of leave-one-out and the following lemma from [4].

**Lemma S-3.** *(Inverse Binomial Lemma from [4]) Given a vector $a := (a_1, \dots, a_m) \in [0,1]^m$, constant $b \in [0,1]$, and independent Bernoulli variables $Z_i \sim Bernoulli(b)$, the weighted sum $Z = 1 + \sum_{i=1}^m a_i Z_i$ satisfies*

$$\frac{1}{1 + b\sum_{i=1}^m a_i} \le \mathbb{E}\left[\frac{1}{Z}\right] \le \frac{1}{b(1 + \sum_{i=1}^m a_i)}. \tag{S-20}$$

We refer reader to the paper for detailed proof of Lemma S-3.

For a fixed $i \in \mathcal{H}_0 \cap \mathcal{S}$, where $\mathcal{S} = \{j : P_j \le \tau, j \in [n]\}$, we use the leave-one-out trick to define some random variable that is independent with $P_i$, say $Y^{-i} := 1 + \sum_{j \in \mathcal{H}_0, j \ne i} \mathbf{1}\{\lambda < P_j \le \tau\}$. In

this way, for all $j \in \mathcal{H}_0, j \neq i, Y_j^{-i} := \mathbf{1}\{\lambda < P_j \leq \tau\}$ is stochastically larger than Bernoulli$(1-\lambda)$ for $j \in \mathcal{H}_0$ conditioning on $P_j \leq \tau$, since the uniformly conservativeness defined in (2) implies that

$$\Pr\{\lambda < P_j \leq \tau \mid P_j \leq \tau\} \geq 1 - \lambda/\tau.$$

Denote $m = |\mathcal{H}_0|$, and $m_S = |\mathcal{H}_0 \cap \mathcal{S}|$, let $Z = 1 + \sum_{i=1}^{m_S - 1} Z_i$, where $\{Z_i\}_{i=1}^{m_S - 1}$ are independent Bernoulli random variables with parameter $1 - \lambda/\tau$. Additionally, since $p$-values are independent of each other, we have

$$\mathbb{E}\left[Y^{-i} \mid \mathcal{S}\right] = 1 + \sum_{j \in \mathcal{H}_0 \cap \mathcal{S}, j \neq i} \mathbb{E}\left[Y_j^{-i} \mid P_j \leq \tau\right]$$
$$\geq 1 + \sum_{j \in \mathcal{H}_0 \cap \mathcal{S}, j \neq i} \mathbb{E}\left[Z_j\right] = \mathbb{E}\left[Z \mid \mathcal{S}\right].$$

Using Lemma S-3, we obtain

$$\mathbb{E}\left[\frac{1}{Y^{-i}} \,\Big|\, \mathcal{S}\right] \leq \mathbb{E}\left[\frac{1}{Z} \,\Big|\, \mathcal{S}\right] \leq \frac{1}{(1 - \lambda/\tau)|\mathcal{H}_0 \cap \mathcal{S}|}. \tag{S-21}$$

Let

$$\widehat{\pi}_0^{-i} := \frac{1 + \sum_{j \leq n, j \neq i} \mathbf{1}\{\lambda < P_j \leq \tau\}}{n(\tau - \lambda)}. \tag{S-22}$$

It is easy to see that $\widehat{\pi}_0^{-i} \geq \frac{Y^{-i}}{n(\tau - \lambda)}$. Together with (S-21) and (S-23), we obtain

$$\mathbb{E}\left[\frac{1}{\widehat{\pi}_0^{-i}} \,\Big|\, \mathcal{S}\right] \leq n(\tau - \lambda)\mathbb{E}\left[\frac{1}{Y^{-i}} \,\Big|\, \mathcal{S}\right] \leq \frac{n\tau}{|\mathcal{H}_0 \cap \mathcal{S}|}. \tag{S-23}$$

Using the definition of $\widehat{\mathrm{FDP}}_{\text{D-StBH}}$ in (S-3), and the uniform conservativeness of $p$-values, we have the following:

$$\mathbb{E}\left[\frac{\sum_{i \in \mathcal{H}_0 \cap \mathcal{S}} \mathbf{1}\{P_i \leq \widehat{s}\}}{(\sum_i \mathbf{1}\{P_i \leq \widehat{s}\}) \vee 1} \,\Big|\, \mathcal{S}\right] \overset{(i)}{\leq} \mathbb{E}\left[\frac{\alpha \sum_{i \in \mathcal{H}_0 \cap \mathcal{S}} \mathbf{1}\{P_i \leq \widehat{s}\}}{n\widehat{\pi}_0 \widehat{s}} \,\Big|\, \mathcal{S}\right]$$

$$= \frac{\alpha}{n}\mathbb{E}\left[\sum_{i \in \mathcal{H}_0 \cap \mathcal{S}} \frac{\mathbf{1}\{P_i \leq \widehat{s}\}}{\widehat{\pi}_0 \widehat{s}} \,\Big|\, \mathcal{S}\right] \overset{(ii)}{=} \frac{\alpha}{n}\mathbb{E}\left[\sum_{i \in \mathcal{H}_0 \cap \mathcal{S}} \frac{\mathbf{1}\{P_i \leq \widehat{s}\}}{\widehat{\pi}_0^{-i}\widehat{s}} \,\Big|\, \mathcal{S}\right]$$

$$\overset{(iii)}{=} \frac{\alpha}{n}\mathbb{E}\left[\sum_{i \in \mathcal{H}_0 \cap \mathcal{S}} \frac{1}{\widehat{\pi}_0^{-i}}\mathbb{E}\left[\frac{\mathbf{1}\{P_i \leq \widehat{s}\}}{\widehat{s}} \,\Big|\, \mathcal{P}^{-i}, \mathcal{S}\right] \,\Big|\, \mathcal{S}\right]$$

$$\overset{(iv)}{=} \frac{\alpha}{n}\mathbb{E}\left[\sum_{i \in \mathcal{H}_0 \cap \mathcal{S}} \frac{1}{\widehat{\pi}_0^{-i}}\mathbb{E}\left[\frac{\mathbf{1}\{P_i \leq \widehat{s}\}}{\widehat{s}} \,\Big|\, \mathcal{P}^{-i}, P_i \leq \tau\right] \,\Big|\, \mathcal{S}\right]$$

$$\overset{(v)}{\leq} \frac{\alpha}{n}\mathbb{E}\left[\sum_{i \in \mathcal{H}_0 \cap \mathcal{S}} \frac{1}{\tau\widehat{\pi}_0^{-i}} \,\Big|\, \mathcal{S}\right] = \frac{\alpha}{n}\sum_{i \in \mathcal{H}_0 \cap \mathcal{S}} \mathbb{E}\left[\frac{1}{\tau\widehat{\pi}_0^{-i}} \,\Big|\, \mathcal{S}\right] \overset{(vi)}{\leq} \alpha,$$

where (i) follows from the condition $\widehat{\mathrm{FDP}}_{\text{D-StBH}} \leq \alpha$; (ii) is true since $\widehat{\pi}_0^{-i} = \widehat{\pi}_0$ given $\mathbf{1}\{P_i \leq \widehat{s}\} = 1$, using the fact that $\widehat{s} \leq \lambda$; (iii) is true since conditioning on $\mathcal{P}_i$ fully determines $\widehat{\pi}_0^{-i}$; (iv) follows from the fact that $\widehat{s} \leq \lambda$; and (v) is obtained by noticing $\widehat{s}$ is coordinatewise nondecreasing in $P_i$ for each $i$, and using the lemma 1 in [4]; and the final step (vi) follows from (S-23). Therefore, we obtain that $\mathbb{E}\left[\mathrm{FDP} \mid \mathcal{S}\right] \leq \alpha$. Taking expectation with regard $\mathcal{S}$ on both side, we have $\mathrm{FDR} \leq \alpha$ as claimed.

## S-9    Proof of Theorem S-3

Theorem S-3 is proved using similar technique in the proof of Theorem 1, we present the proof here for completeness. Similarly, we need the following lemma for the proof, which is proved in Section S-9.1.

**Lemma S-4.** *Assume that the p-values $P_1, P_2, \ldots$ are independent and let $g : \{0,1\}^T \to \mathbb{R}$ be any coordinatewise nondecreasing function. Further, assume that $\alpha_t, \lambda_t$ and $1 - \tau_t$ are all monotonic functions of the past as defined in Section S-3, while satisfying the constraints $\alpha_t \leq \lambda_t < \tau_t$ for all t. Then, for any index $t \leq T$ such that $H_t \in \mathcal{H}_0$, we have:*

$$\mathbb{E}\left[\frac{\alpha_t \mathbf{1}\{\lambda_t < P_t \leq \tau_t\}}{(\tau_t - \lambda_t)(g(|\mathcal{R}|_{1:T}) \vee 1)} \;\middle|\; \mathcal{F}^{E_t - 1}, S_t = 1\right] \geq \mathbb{E}\left[\frac{\alpha_t}{\tau_t(g(|\mathcal{R}|_{1:T}) \vee 1)} \;\middle|\; \mathcal{F}^{E_t - 1}, S_t = 1\right]$$

$$\geq \mathbb{E}\left[\frac{\mathbf{1}\{P_t \leq \alpha_t\}}{g(|\mathcal{R}|_{1:T}) \vee 1} \;\middle|\; \mathcal{F}^{E_t - 1}, S_t = 1\right],$$

*where $|\mathcal{R}|_{1:T} := \{|\mathcal{R}_1|, \ldots, |\mathcal{R}_T|\}$.*

Denote $\mathcal{R}(t) = \{i : P_i \leq \alpha_i, E_i \leq t\}$, we have the following:

$$\mathbb{E}\left[|\mathcal{H}_0 \cap \mathcal{R}(t)|\right] = \mathbb{E}\left[\sum_{E_j \leq t, j \in \mathcal{H}_0} \mathbf{1}\{P_j \leq \alpha_j\}\right] \overset{(i)}{\leq} \sum_{j \leq t, j \in \mathcal{H}_0} \mathbb{E}\left[\mathbf{1}\{P_j \leq \alpha_j\}\right]$$

$$\overset{(ii)}{=} \sum_{j \leq t, j \in \mathcal{H}_0} \mathbb{E}\left[\mathbb{E}\left[\mathbf{1}\{P_j \leq \alpha_j\} \;\middle|\; \mathcal{F}^{E_j - 1}, S_j\right]\right]$$

$$\overset{(iii)}{=} \sum_{j \leq t, j \in \mathcal{H}_0} \mathbb{E}\left[\mathbb{E}\left[\mathbf{1}\{\frac{P_j}{\tau_j} \leq \frac{\alpha_j}{\tau_j}\} \;\middle|\; P_j \leq \tau_j, \mathcal{F}^{E_j - 1}\right]\Pr\{S_j = 1 \mid \mathcal{F}^{E_j - 1}\}\right]$$

$$\overset{(iv)}{\leq} \sum_{j \leq t, j \in \mathcal{H}_0} \mathbb{E}\left[\frac{\alpha_j}{\tau_j}\Pr\{S_j = 1 \mid \mathcal{F}^{E_j - 1}\}\right],$$

where step (i) is true since the set of rejections by time $t$ could be at most $[t]$; and (ii) is obtained via taking iterated expectation by conditioning on $\{\mathcal{F}^{E_j - 1}, S_j\}$ respectively for each $j \in \mathcal{H}_0$; and (iii) is true since $\alpha_j \leq \tau_j$; and finally, step (iv) follows from the uniformly conservativeness of nulls. Next, notice that

$$\sum_{j \leq t, j \in \mathcal{H}_0} \mathbb{E}\left[\frac{\alpha_j}{\tau_j}\Pr\{S_j = 1 \mid \mathcal{F}^{E_j - 1}\}\right]$$

$$\overset{(v)}{\leq} \sum_{j \leq t, j \in \mathcal{H}_0} \mathbb{E}\left[\frac{\alpha_j}{\tau_j}\mathbb{E}\left[\frac{\mathbf{1}\{\lambda_j < P_j\}}{(1 - \lambda_j/\tau_j)} \;\middle|\; P_j \leq \tau_j, \mathcal{F}^{E_j - 1}\right]\Pr\{S_j = 1 \mid \mathcal{F}^{E_j - 1}\}\right]$$

$$= \sum_{j \leq t, j \in \mathcal{H}_0} \mathbb{E}\left[\frac{\alpha_j}{\tau_j}\mathbb{E}\left[\frac{\mathbf{1}\{\lambda_j < P_j \leq \tau_j\}}{(1 - \lambda_j/\tau_j)} \;\middle|\; S_j, \mathcal{F}^{E_j - 1}\right]\right]$$

$$= \sum_{j \leq t, j \in \mathcal{H}_0} \mathbb{E}\left[\alpha_j \frac{\mathbf{1}\{\lambda_j < P_j \leq \tau_j\}}{(\tau_j - \lambda_j)}\right]$$

where (v) is true because of the uniformly conservativaness of null $p$-values, and the last two equalities use the predictability of $\alpha_j$ and $\tau_j$ with regard $\mathcal{F}^{E_j - 1}$. Then, by removing some constrains on the index, and applying the condition that $\widehat{\text{FDP}}_{\text{ADDIS}_{\text{async}}} \leq \alpha$, one obatin

$$\sum_{j \leq t, j \in \mathcal{H}_0} \mathbb{E}\left[\alpha_j \frac{\mathbf{1}\{\lambda_j < P_j \leq \tau_j\}}{(\tau_j - \lambda_j)}\right]$$

$$\leq \sum_{j \leq t} \mathbb{E}\left[\frac{\alpha_j}{(\tau_j - \lambda_j)}(\mathbf{1}\{\lambda_j < P_j \leq \tau_j, E_j < t\} + \mathbf{1}\{E_j \geq t\})\right]$$

$$\leq \alpha \, \mathbb{E}\left[\left(\sum_{j \leq t}\mathbf{1}\{P_j \leq \alpha_j, E_j < t\}\right) \vee 1\right] = \alpha \, \mathbb{E}\left[|\mathcal{R}(t)| \vee 1\right],$$

Therefore, we have

$$\mathbb{E}\left[|\mathcal{H}_0 \cap \mathcal{R}(t)|\right] \leq \alpha \mathbb{E}\left[|\mathcal{R}(t)| \vee 1\right].$$

After rearranging the terms above, we have $\text{mFDR}(t) \leq \alpha$, as claimed. Therefore, we finished the proof of first part of Theorem S-3.

Using the same tricks of taking iterated expectation, we have the following:

$$
\begin{aligned}
\text{FDR}(t) = \mathbb{E}\left[\text{FDP}(t)\right] = \mathbb{E}\left[\frac{|\mathcal{H}_0 \cap \mathcal{R}(t)|}{|\mathcal{R}(t)| \vee 1}\right] &= \mathbb{E}\left[\frac{\sum_{E_j \leq t, j \in \mathcal{H}_0} \mathbf{1}\{P_j \leq \alpha_j\}}{|\mathcal{R}(t)| \vee 1}\right] \\
&\leq \mathbb{E}\left[\frac{\sum_{j \leq t, j \in \mathcal{H}_0} \mathbf{1}\{P_j \leq \alpha_j\}}{|\mathcal{R}(t)| \vee 1}\right] = \sum_{j \leq t, j \in \mathcal{H}_0} \mathbb{E}\left[\frac{\mathbf{1}\{P_j \leq \alpha_j\}}{|\mathcal{R}(t)| \vee 1}\right] \quad \text{(S-24)} \\
&= \sum_{j \leq t, j \in \mathcal{H}_0} \mathbb{E}\left[\mathbb{E}\left[\frac{\mathbf{1}\{P_j \leq \alpha_j\}}{|\mathcal{R}(t)| \vee 1} \,\middle|\, S_j, \mathcal{F}^{E_j - 1}\right]\right].
\end{aligned}
$$

Under additional assumptions about the independence of $p$-values and monotonicity of $\alpha_t$ and $\lambda_t$ for each $t \in N$, and notice that $|\mathcal{R}(t)| = \sum_{i \leq t, E_i < t} R_i = \sum_{i < t} |\mathcal{R}_i|$ is coordinatewise nondecreasing function of $|\mathcal{R}|_{1:t}$, we apply Lemma S-4 to the RHS of (S-24) to obtain the following:

$$
\begin{aligned}
&\sum_{j \leq t, j \in \mathcal{H}_0} \mathbb{E}\left[\mathbb{E}\left[\frac{\mathbf{1}\{P_j \leq \alpha_j\}}{|\mathcal{R}(t)| \vee 1} \,\middle|\, S_j, \mathcal{F}^{E_j - 1}\right]\right] \\
&= \sum_{j \leq t, j \in \mathcal{H}_0} \mathbb{E}\left[\mathbb{E}\left[\frac{\mathbf{1}\{P_j \leq \alpha_j\}}{|\mathcal{R}(t)| \vee 1} \,\middle|\, S_j = 1, \mathcal{F}^{E_j - 1}\right] \Pr\{S_j = 1 \mid \mathcal{F}^{E_j - 1}\}\right] \\
&\leq \sum_{j \leq t, j \in \mathcal{H}_0} \mathbb{E}\left[\mathbb{E}\left[\frac{\alpha_j}{\tau_j(|\mathcal{R}(t)| \vee 1)} \,\middle|\, S_j = 1, \mathcal{F}^{E_j - 1}\right] \Pr\{S_j = 1 \mid \mathcal{F}^{E_j - 1}\}\right] \\
&\leq \sum_{j \leq t, j \in \mathcal{H}_0} \mathbb{E}\left[\mathbb{E}\left[\frac{\alpha_j}{\tau_j(|\mathcal{R}(t)| \vee 1)} \frac{\mathbf{1}\{\lambda_j < P_j \leq \tau_j\}}{1 - \lambda_j/\tau_j} \,\middle|\, S_j = 1, \mathcal{F}^{E_j - 1}\right] \Pr\{S_j = 1 \mid \mathcal{F}^{E_j - 1}\}\right]
\end{aligned}
$$

(S-25)

Once again using the fact that $\alpha_j \leq \tau_j$ for all $j$, and the law of iterated expectation, the RHS of (S-25) equals

$$
\begin{aligned}
&\sum_{j \leq t, j \in \mathcal{H}_0} \mathbb{E}\left[\mathbb{E}\left[\frac{\alpha_j}{\tau_j(|\mathcal{R}(t)| \vee 1)} \frac{\mathbf{1}\{\lambda_j < P_j \leq \tau_j\}}{1 - \lambda_j/\tau_j} \,\middle|\, S_j, \mathcal{F}^{E_j - 1}\right]\right] \\
&= \sum_{j \leq t, j \in \mathcal{H}_0} \mathbb{E}\left[\frac{\alpha_j}{\tau_j(|\mathcal{R}(t)| \vee 1)} \frac{\mathbf{1}\{\lambda_j < P_j \leq \tau_j\}}{(1 - \lambda_j/\tau_j)}\right] \\
&\leq \sum_{j \leq t} \mathbb{E}\left[\frac{1}{|\mathcal{R}(t)| \vee 1} \frac{\alpha_j}{(\tau_j - \lambda_j)}(\mathbf{1}\{\lambda_j < P_j \leq \tau_j, E_j < t\} + \mathbf{1}\{E_j \geq t\})\right] \\
&= \mathbb{E}\left[\widehat{\text{FDP}}_{\text{ADDIS}_{\text{async}}}(t)\right] \leq \alpha.
\end{aligned}
$$

(S-26)

Therefore, combining (S-24), (S-26), and (S-26), we conclude $\text{FDR}(t) \leq \alpha$. This finishes the proof of the second part of the theorem.

### S-9.1  Proof of Lemma S-4

Similar to the proof of Lemma S-1, we prove this lemma by constructing a hallucinated vector. Specifically, to prove the first part of the inequality, for any fixed $t \in \mathbb{N}$, for all $i \in \mathbb{N}$, let $\widetilde{P}_i = \tau_i \cdot \mathbf{1}\{i = t\} + P_i \cdot \mathbf{1}\{i \neq t\}$, and keep the finish times for all the tests unchanged. Then we denote the testing levels, candidate levels and selected levels resulted from the hallucinated $\{\widetilde{P}_i\}$ as $\{\widetilde{\alpha}_i\}$, $\{\widetilde{\lambda}_i\}$ and $\{\widetilde{\tau}_i\}$ respectively. Correspondingly, we let

$$
\widetilde{S}_i = \mathbf{1}\{\widetilde{P}_i \leq \widetilde{\tau}_i\}, \quad \widetilde{C}_i = \mathbf{1}\{\widetilde{P}_i \leq \widetilde{\lambda}_i\}, \quad \widetilde{R}_i = \mathbf{1}\{\widetilde{P}_i \leq \widetilde{\alpha}_i\}.
$$

Given $\lambda_t < P_t \leq \tau_t$, we have $\widetilde{S}_t = S_t = 1, \widetilde{R}_t = R_t = 0, \widetilde{C}_t = C_t = 0$. This implies $\mathcal{R}_{1:T} = \widetilde{\mathcal{R}}_{1:T}$. We then obtain the following:

$$\mathbb{E}\left[\frac{\alpha_t \mathbf{1}\{\lambda_t < P_t \leq \tau_t\}}{(\tau_t - \lambda_t)(g(|\mathcal{R}|_{1:T}) \vee 1)} \,\bigg|\, S_t = 1, \mathcal{F}^{E_t - 1}\right] = \mathbb{E}\left[\frac{\alpha_t \mathbf{1}\{\lambda_t < P_t \leq \tau_t\}}{(\tau_t - \lambda_t)(g(|\widetilde{\mathcal{R}}|_{1:T}) \vee 1)} \,\bigg|\, S_t = 1, \mathcal{F}^{E_t - 1}\right]$$

$$\overset{(i)}{=} \mathbb{E}\left[\frac{\alpha_t}{\tau_t(g(|\widetilde{\mathcal{R}}|_{1:T}) \vee 1)} \,\bigg|\, S_t = 1, \mathcal{F}^{E_t - 1}\right] \mathbb{E}\left[\frac{\mathbf{1}\{\lambda_t < P_t \leq \tau_t\}}{(1 - \lambda_t/\tau_t)(g(|\widetilde{\mathcal{R}}|_{1:T}) \vee 1)} \,\bigg|\, S_t = 1, \mathcal{F}^{E_t - 1}\right]$$

$$\overset{(ii)}{\geq} \mathbb{E}\left[\frac{\alpha_t}{\tau_t(g(|\widetilde{\mathcal{R}}|_{1:T}) \vee 1)} \,\bigg|\, S_t = 1, \mathcal{F}^{E_t - 1}\right] \overset{(iii)}{\geq} \mathbb{E}\left[\frac{\alpha_t}{\tau_t(g(|\mathcal{R}|_{1:T}) \vee 1)} \,\bigg|\, S_t = 1, \mathcal{F}^{E_t - 1}\right],$$

where (i) is obtained from the fact that $\widetilde{\mathcal{R}}_{1:T}$ is independent of $P_t$, and (ii) is true because of the uniformly conservativeness of null $p$-values, and (iii) is true since $\widetilde{R}_i \subseteq R_i$ for all $i$ given $S_t = 1$ using the similar logic in the proof of Lemma S-1 in Section S-5.1, that is utilizing the monotonicity assumptions of $\alpha_t$, $\lambda_t$, and $\tau_t$.

Similarly, for the second part of the inequality, we construct $\widetilde{P}_i = P_i \cdot \mathbf{1}\{i = t\}$, and keep the finish times for all the tests unchanged, while we define $\widetilde{\alpha}_t, \widetilde{\lambda}_t, \widetilde{\tau}_t$ and $\widetilde{R}_t, \widetilde{C}_t, \widetilde{S}_t$ in the same way as in the proof of the first part. Notice that $\widetilde{\mathcal{R}}_{1:T}$ is independent of $P_t$, and that given $P_t \leq \alpha_t$, we have $\widetilde{S}_t = S_t = 1, \widetilde{R}_t = \widetilde{C}_t = R_t = C_t = 1$, which leads to $\mathcal{R}_{1:T} = \widetilde{\mathcal{R}}_{1:T}$. Then we have the following:

$$\mathbb{E}\left[\frac{\mathbf{1}\{P_t \leq \alpha_t\}}{g(|\mathcal{R}|_{1:T}) \vee 1} \,\bigg|\, S_t = 1, \mathcal{F}^{E_t - 1}\right] = \mathbb{E}\left[\frac{\mathbf{1}\{P_t \leq \alpha_t\}}{g(|\widetilde{\mathcal{R}}|_{1:T}) \vee 1} \,\bigg|\, S_t = 1, \mathcal{F}^{E_t - 1}\right]$$

$$\leq \mathbb{E}\left[\frac{\alpha_t}{\tau_t(g(|\widetilde{\mathcal{R}}|_{1:T}) \vee 1)} \,\bigg|\, S_t = 1, \mathcal{F}^{E_t - 1}\right] \leq \mathbb{E}\left[\frac{\alpha_t}{\tau_t(g(|\mathcal{R}|_{1:T}) \vee 1)} \,\bigg|\, S_t = 1, \mathcal{F}^{E_t - 1}\right].$$

This concludes the whole proof Lemma S-4.

## S-10    An equivalent form of ADDIS$^*$ algorithm

---

**Algorithm S-3:** The ADDIS$^*$ algorithm with explicit use of discarding

---

**Input:** FDR level $\alpha$, discarding threshold $\tau \in (0, 1]$, candidate threshold $\lambda \in [\alpha, \tau)$, sequence
$\{\gamma_j\}_{j=0}^{\infty}$ which is nonnegative, nonincreasing and sums to one, initial wealth $W_0 \leq \alpha$.

**for** $t = 1, 2, \ldots$ **do**

    **if** $P_t > \tau$ **then**

        | Discard $P_t$ and move to next round.

    **end**

    **else**

        Reject the $t$-th null hypothesis if $P_t/\tau \leq \alpha_t$, where

        $\alpha_t := (1 - \lambda)\left(W_0\gamma_{S^t - C_{0+}} + (\alpha - W_0)\gamma_{S^t - \kappa_1^* - C_{1+}} + \alpha\sum_{j \geq 2}\gamma_{S^t - \kappa_j^* - C_{j+}}\right).$

        Here,  $S^t = \sum_{i < t}\mathbf{1}\{P_i \leq \tau\},\quad C_{j+} = \sum_{i = \kappa_j + 1}^{t-1}\mathbf{1}\{P_i \leq \lambda\},$

            $\kappa_j = \min\{i \in [t-1] : \sum_{k \leq i}\mathbf{1}\{P_k \leq \alpha_k\} \geq j\},\quad \kappa_j^* = \sum_{i \leq \kappa_j}\mathbf{1}\{P_i \leq \tau\}.$

    **end**

**end**

---

## S-11    Heatmap of $g \circ F$

Here we show the heatmap of $g \circ F$ versus $\theta := \lambda/\tau$ and $\tau$ given different choices of $F$. Specifically, we let $F$ be the CDF of all $p$-values (nulls and alternatives taken together) drawn as described in Section 3, with different choices of $\mu_N, \mu_A$, and $\pi_A$. In Figure S-1, we show results for $\mu_N \in \{-0.5, -1\}$, $\mu_A \in \{2, 3\}$, and $\pi_A \in \{0.2, 0.3\}$ respectively, which are some reasonably common

settings that one may expect in practice. We see that the heatmap of $g \circ F$ demonstrates the same consistent pattern across different choices of $F$.

Figure S-1: The heatmap of function $g \circ F$, where $F$ is the CDF of $p$-values drawn as described in Section 3 with $\mu_N = -0.5, \mu_A = 2, \pi_A = 0.2$ for plot (a); $\mu_N = -0.5, \mu_A = 3, \pi_A = 0.2$ for plot (b); $\mu_N = -1, \mu_A = 2, \pi_A = 0.2$ for plot (c); $\mu_N = -1, \mu_A = 3, \pi_A = 0.2$ for plot (d); $\mu_N = -0.5, \mu_A = 2, \pi_A = 0.3$ for plot (e); $\mu_N = -0.5, \mu_A = 3, \pi_A = 0.3$ for plot (f); $\mu_N = -1, \mu_A = 2, \pi_A = 0.3$ for plot (g); $\mu_N = -1, \mu_A = 3, \pi_A = 0.3$ for plot (h).