[Reviews · NeurIPS 2019]

Reviewer 1



The manuscript if very well written. The problem of "error-guarantees on a stream of decisions" is of great interest. Conservative p-values mean that the data "surprised" the researcher. In my experience, this rarely happens. It seems, however, that the price of protection from such a scenario is not high. This, in my view, is the best feature of ADDIS.

Reviewer 2



The paper is very well written. It’s a pleasure to read. Online FDR control for conservative nulls is an important problem and the method is novel. The numerical studies show superior power performance compared to existing methods when the null p-values are conservative. However there is no real-data experiments in the paper. It would be interesting to see if the new method can give better discoveries in practical problems. Minor points: Line 120: FDP_LORD++ -> \hat{FDP}_LORD++ Line 158: Do you mean \lambda*\tau <= \alpha or > \alpha?

Reviewer 3



Overall, I think this is a good paper. Originality: This paper combines the work of two previous algorithms. Yet such combination does not seem to be trivial. So I believe the work is somewhat novel. Quality: The proof looks right. One comment is that the paper claims that convex CDF function would satisfy the condition of equation (3). CDF is bounded and therefore can not be a convex function on R if there is no further clarification. And some concrete examples of this claim would be nice. Clarity: It is easy to follow and understand the motivation. Significance: The paper gives a hybrid of the previous two algorithms LORD and SAFFRON and empirically shows that the new algorithm has inherited the advantages of both algorithms. In my opinion, it is interesting. However, I could be wrong since I am not an expert in this field of literature.

[Author Response · NeurIPS 2019]

We thank the reviewers for their positive comments. Below we address some minor concerns raised.

• Reviewer 1: Thank you for supporting our work. One minor clarification:

[...*conservative p-values mean the data "surprised" the researcher...*] We expect conservative nulls to be the norm
in practice primarily because they would be uniform only if the parameter lay at the boundary of the null set. For
example, in an A/B test the null is: $\theta_B \leq \theta_A$, and the p-value would be uniform only if $\theta_B = \theta_A$, and would be
conservative otherwise (as explained in lines 58-67 of the paper). We can add an explicit example. As you rightly
point out, the main strength of ADDIS is that the price of protection is minor.

• Reviewer 2: Thank you for your careful reading. Indeed, it should be $\widehat{\mathrm{FDP}}_{\mathrm{LORD++}}$ in the line 120, and $\lambda\tau > \alpha$ in line
158. We will correct these typos.

[...*demonstrate with real-data experiments...*] The request about the real data experiments is fair enough. Unfortu-
nately, there are two hurdles with real data. The first is that we are not aware of any publicly available dataset of
this kind, because (for example) all the tech companies that run such large sequences of experiments keep their data
proprietary. The second is that even if the data were made available and ADDIS had more discoveries than (say)
LORD++, we would not know which of the extra discoveries were false positives, and which were true positives,
because the ground truth is unknown. Thus we resort to simulations to compare algorithms, which are realistic since
practitioners often resort to the central limit theorem (averages behave like Gaussians) to design their tests (like
t-tests). We will clarify this in the paper.

• Reviewer 4: Thank you for your thoughtful review.

[...*CDF is bounded and therefore can not be a convex function on R...*] Indeed, a CDF defined on $\mathbb{R}$ cannot be convex.
However, the argument is made for the CDF of $p$-values, which is defined in the range of $[0, 1]$. Concrete examples
for $p$-values with convex CDF involve the content following the two bullet points in lines 53-56. We will clarify this.

[Meta-Review · NeurIPS 2019]

The reviewers are all positive about the contributions, novelty and clarity of the paper.